# Neural network emulation of the human ventricular cardiomyocyte action potential for more efficient computations in pharmacological studies

Thomas Grandits[1,2]\*, Christoph M Augustin[3,4], Gundolf Haase[1], Norbert Jost[5,6], Gary R Mirams[7], Steven A Niederer[8], Gernot Plank[3,4], András Varró[5,6], László Virág[5], Alexander Jung[3]\*

[1]Department of Mathematics and Scientific Computing, University of Graz, Graz, Austria; [2]NAWI Graz, University of Graz, Graz, Austria; [3]Gottfried Schatz Research Center for Cell Signaling, Metabolism and Aging - Division of Medical Physics and Biophysics, Medical University of Graz, Graz, Austria; [4]BioTechMed-Graz, Graz, Austria; [5]Department of Pharmacology and Pharmacotherapy, University of Szeged, Szeged, Hungary; [6]HUN-REN-TKI, Research Group of Pharmacology, Budapest, Hungary; [7]Centre for Mathematical Medicine & Biology, School of Mathematical Sciences, University of Nottingham, Nottingham, United Kingdom; [8]Division of Imaging Sciences & Biomedical Engineering, King's College London, London, United Kingdom

\*For correspondence:
thomas.grandits@uni-graz.at (TG);
alexander.jung@medunigraz.at (AJ)

**Competing interest:** The authors declare that no competing interests exist.

**Abstract** Computer models of the human ventricular cardiomyocyte action potential (AP) have reached a level of detail and maturity that has led to an increasing number of applications in the pharmaceutical sector. However, interfacing the models with experimental data can become a significant computational burden. To mitigate the computational burden, the present study introduces a neural network (NN) that emulates the AP for given maximum conductances of selected ion channels, pumps, and exchangers. Its applicability in pharmacological studies was tested on synthetic and experimental data. The NN emulator potentially enables massive speed-ups compared to regular simulations and the forward problem (find drugged AP for pharmacological parameters defined as scaling factors of control maximum conductances) on synthetic data could be solved with average root-mean-square errors (RMSE) of 0.47 mV in normal APs and of 14.5 mV in abnormal APs exhibiting early afterdepolarizations (72.5% of the emulated APs were alining with the abnormality, and the substantial majority of the remaining APs demonstrated pronounced proximity). This demonstrates not only very fast and mostly very accurate AP emulations but also the capability of accounting for discontinuities, a major advantage over existing emulation strategies. Furthermore, the inverse problem (find pharmacological parameters for control and drugged APs through optimization) on synthetic data could be solved with high accuracy shown by a maximum RMSE of 0.22 in the estimated pharmacological parameters. However, notable mismatches were observed between pharmacological parameters estimated from experimental data and distributions obtained from the Comprehensive in vitro Proarrhythmia Assay initiative. This reveals larger inaccuracies which can be attributed particularly to the fact that small tissue preparations were studied while the emulator was trained on single cardiomyocyte data. Overall, our study highlights the potential of NN emulators as powerful tool for an increased efficiency in future quantitative systems pharmacology studies.

## eLife assessment

This **valuable** prospective study develops a new tool to accelerate pharmacological studies by using neural networks to emulate the human ventricular cardiomyocyte action potential. The evidence supporting the conclusions is **convincing**, based on using a large and high-quality dataset to train the neural network emulator. There are nevertheless a few areas in which the article may be improved through validating the neural network emulators against extensive experimental data. In addition, the article may be improved through delineating the exact speed-up achieved and the scope for acceleration. The work will be of broad interest to scientists working in cardiac simulation and quantitative system pharmacology.

## Introduction

Computer models of human physiology are becoming increasingly detailed and mature and the area of ventricular cardiomyocyte electrophysiology (EP) is one of the most advanced. The most updated models include fine representations of ion movements through various important channels, pumps, and exchangers, and take the complex handling of intracellular calcium accurately into account (*Grandi et al., 2010*; *O'Hara et al., 2011*; *Tomek et al., 2019*; *Bartolucci et al., 2020*). While these models have individual strengths and limitations in replicating different aspects of physiology, pathology, and pharmacology (*Corrado et al., 2021*; *Amuzescu et al., 2021*), their degree of credibility has reached a level that has led to an increasing number of applications in academia and beyond. This holds in particular for the pharmaceutical sector, where much effort is spent on using computer modeling to reduce traditional preclinical and clinical methodologies for assessing the efficacy and safety of novel drug candidates (*Mirams et al., 2011*; *Passini et al., 2017*; *Li et al., 2019*; *Passini et al., 2021*). To improve the regulatory assessment of a drug's proarrhythmic potential, the Comprehensive in Vitro Proarrhythmia Assay (CiPA) was proposed in 2013 following a workshop at the US Food and Drug Administration (*Sager et al., 2014*; *Colatsky et al., 2016*; *Strauss et al., 2021*). A central component is a computer model of human ventricular cardiomyocyte EP that is coupled to a pharmacological model describing the interaction between a given drug and multiple arrhythmia-relevant channels (*Dutta et al., 2017*; *Li et al., 2017*; *Li et al., 2019*). For a given drug, experimental channel block data are collected to inform the pharmacological model and corresponding simulations of the action potentials (AP; time course of the transmembrane potential) are performed to predict the proarrhythmic risk based on a mechanistically motivated biomarker (*Chang et al., 2017*; *Li et al., 2017*; *Li et al., 2019*). The prediction is then compared with findings in experimental (*Blinova et al., 2018*) and clinical (*Vicente et al., 2018*) studies. To compute the drugged AP for given pharmacological parameters is a forward problem, while the corresponding inverse problem is to find pharmacological parameters for given control (before drug administration) and drugged AP. Some relevant examples for the latter have been presented by *Bottino et al., 2006* who estimated pharmacological parameters from APs of canine Purkinje fibers and by Tv*eito* et al. *who estima*ted pharmacological parameters from AP biomarkers measured in human induced pluripotent stem cell-derived cardiomyocytes (*Tveito et al., 2018*) and several animal ventricular cardiomyocytes (*Tveito et al., 2020*). Furthermore, *Jaeger et al., 2021* identified the optimal polypharmacological treatment for recovering APs of mutant ventricular cardiomyocytes based on biomarkers of simulated wild type and mutant APs.

When the models are interfaced with experimental data, attention should be paid to the inherent uncertainty in the data that results from beat-to-beat variability (intrinsic variability), cell-to-cell variability (extrinsic variability), and measurement errors (observational uncertainty) (*Mirams et al., 2016*). Uncertainty propagates through the given problem from APs to estimated parameters or from parameters to predicted APs and must be properly quantified to draw reliable conclusions from the results. Multiple methodologies exist for this purpose (*Oakley and O'Hagan, 2004*; *Mirams et al., 2016*; *Sher et al., 2022*) but usually require many simulations, which even for ordinary differential equation (ODE)-based models of cardiomyocyte EP can become a significant computational burden when considering that each simulation includes a substantial number of beats to reach the model's limit cycle, (also often called steady state). To overcome this problem, surrogate models have emerged which approximate (emulate) chosen outputs for given inputs multiple orders of magnitude faster. In line with uncertainty quantification literature, the cardiomyocyte EP model is from now on termed the '*simulator*', whereas the surrogate model is termed the '*emulator*'. Earlier work has reported on an

emulator based on linear interpolation of a multi-dimensional lookup table *Mirams et al., 2014* and more recently, Gaussian process (GP) emulators have become popular. Their key advantage is that in- and outputs are modeled as random distributions which allows for rapid sampling of the posterior distributions (*Chang et al., 2015*; *Johnstone et al., 2016*; *Coveney and Clayton, 2018*; *Ghosh, 2018*; *Rasmussen, 2019*; *Coveney et al., 2021*) and while outputs of recently published GP emulators were relevant biomarkers of the AP (*Chang et al., 2015*; *Johnstone et al., 2016*; *Coveney and Clayton, 2018*; *Ghosh, 2018*; *Coveney et al., 2021*), the emulation of the entire AP can also be realized, for example through dimensionality reduction techniques such as the principal component analysis or regressing state-transition models (*Mohammadi et al., 2019*). However, GP emulators are not well suited to capture discontinuities of the response surface as standard GP emulators assume a smooth and continuous response to changes in parameter values. Applying GPs for modeling discontinuous functions therefore remains a largely open problem. Thus, they may fail to capture AP abnormalities, which is a particular drawback for pharmacological studies where bifurcations in behavior such as early afterdepolarizations (EAD) can occur (*Ghosh, 2018*). To address this, *Ghosh, 2018* presented a two-step approach for the emulation of the AP duration at 90% repolarization that first sets up a GP for the location of discontinuities and then fits separate GP emulators for the output of interest either side of these boundaries. In contrast, it has been proven that neural networks (NN) can approximate even discontinuous functions with arbitrary precision in theory (*Hornik et al., 1989*), while recent works using NNs show empirically promising results for modeling partial differential equations containing discontinuities (*Jagtap et al., 2020*). These features render NN emulators suitable emulation candidates and while *Lei and Mirams, 2021* have recently investigated NN emulation of hERG channel kinetics, *Jeong et al., 2023* proposed a neural network using AP shapes as input for the prediction of a drug's proarrhythmic risk. However, to the best of our knowledge, NN emulators have not yet been used as surrogate for cardiomyocyte EP models.

The present study introduces NN emulation of the human ventricular cardiomyocyte AP and investigates the applicability in pharmacological studies. To this end, a NN emulator was developed based on data generated using a state-of-the-art simulator (*Tomek et al., 2019*; *Tomek et al., 2020*) and the evaluation was done for forward and inverse problems on synthetic and experimental data.

## Materials and methods

The methodology of this study including the development of the emulator and the evaluation is outlined in *Figure 1*.

### Simulator

The simulator of *Tomek et al., 2020* (ToR-ORd-dynCl simulator) was used. This is available in CARPentry (*Vigmond et al., 2008*) and was implemented based on the published CellML file for the endocardial subtype (ToRORd_dynCl_endo.cellml, https://github.com/jtmff/torord/tree/master/cellml; *jtmff, 2020*). Simulations were performed in CARPentry with the single cell tool bench. To compute the gating variables, the Rush-Larsen Method (*Rush and Larsen, 1978*) was employed, which uses an analytical solution assuming fixed voltage over a small timestep, and the remaining variables were computed by the Forward Euler method. To ensure low computational cost, we found the maximum solver and sampling time steps that still produce accurate results as follows. Various solver and sampling time steps were applied to generate APs and the biomarkers (AP biomarkers and abnormalities) used in this study were computed and compared with those that correspond to the minimum time steps (solver: 0.005 ms; sampling: 0.01 ms). We considered the time steps with only 2% relative difference for all AP biomarkers (solver: 0.01 ms; sampling: 0.05 ms) to offer a sufficiently good approximation. APs were stimulated at a pacing cycle length of 1000 ms using 1 ms long rectangular transmembrane current density pulses of 53 $\frac{\mu A}{cm^2}$ (*Tomek et al., 2019*). To approach the simulator's steady state, a series of 1000 stimuli were applied for each new parameter set starting from the initial states specified in the CellML file (when 1000 additional stimuli were applied, the maximum intracellular [$Ca^{2+}$], [$Cl^-$], and [$Na^+$] changed by 1.5%, 0.15%, and 1.7%, respectively). Note, that the simulations can also be performed using open-source software such as (*Clerx et al., 2016*), OpenCOR (*Garny and Hunter, 2015*) and openCARP (*Plank et al., 2021*).

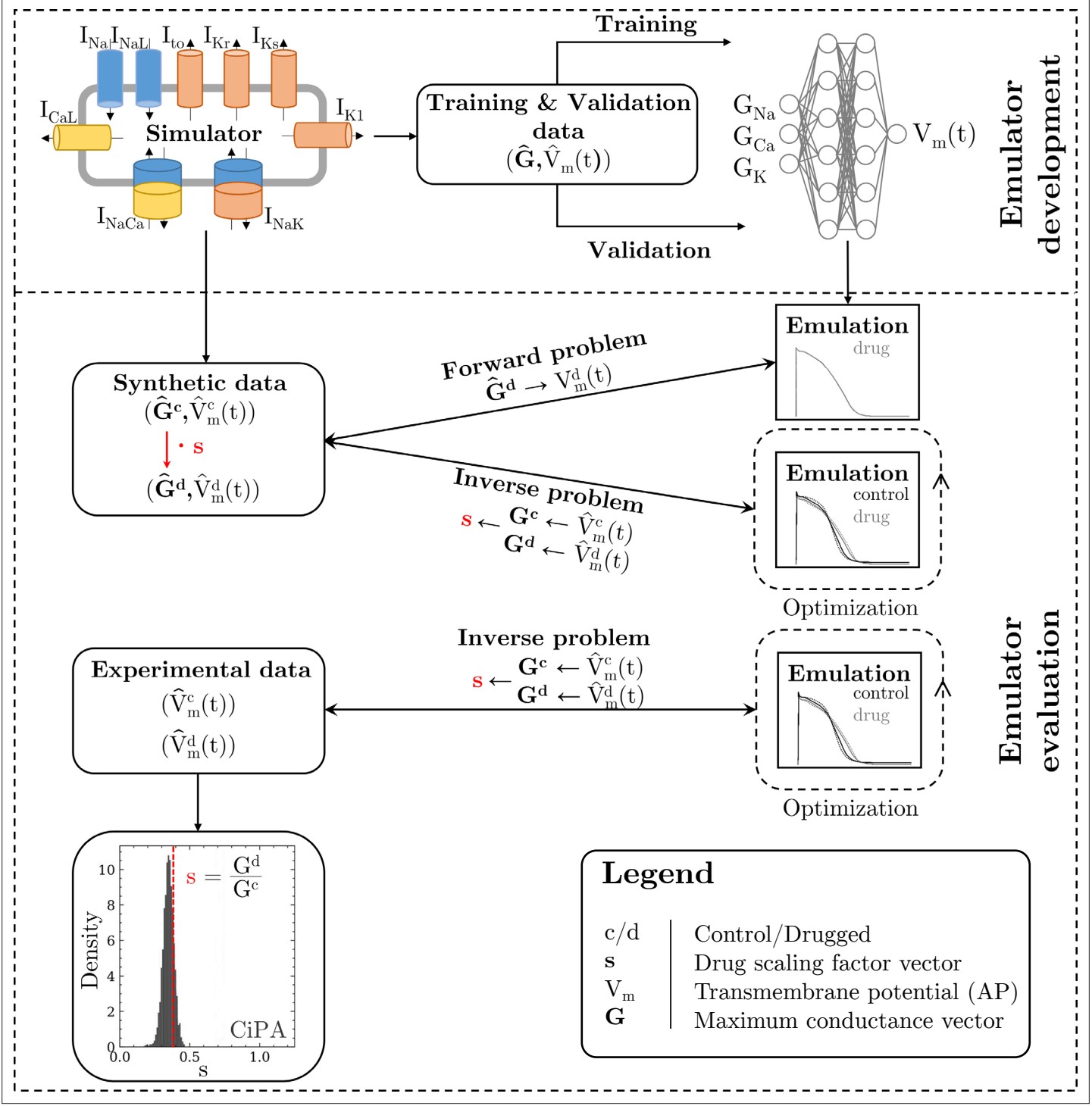

**Figure 1.** Methodology of this study including the emulator development and the evaluation. The simulator is visualized by a schematic human ventricular cardiomyocyte that includes all currents considered for the emulator training. Inputs of the emulator (see **Figure 3**) are the corresponding maximum conductances (*G*) but for the sake of illustration, only three representatives are shown. Output is the AP (*V$_m$(t)*). Training and validation data (maximum conductances $\hat{G}$ and APs $\hat{V}_m(t)$) were generated by the simulator. The evaluationwas performed for forward and inverse problems and to this end, the pharmacological parameter *s* was introduced. This describes the interaction between the drug and a given target and was defined as scaling factor of the respective maximum conductance in control conditions (s < 1:block, s=1:no effect, s > 1:enhancement). Synthetic data (control maximum conductances and drugged maximum conductances obtained through scaling, and control and drugged APs) and experimental data (control and drugged APs) were used for the evaluation (***Orvos, 2019***). The forward problem was only solved for whereas the inverse problem was solved for both synthetic and experimental data. When experimental data were used, estimated pharmacological parameters were compared to distributions derived from data published within the CiPA initiative (***Chang et al., 2017***; ***Li et al., 2017***).

## Emulator

Here, we present the architecture and capabilities of the implemented and trained emulator. Details on the practical implementation and a link to the public code can be found in Code & data availability.

### Data

We generated three data sets in the study to train, validate and test the emulator performance.

#### Training/validation data (#1)

The first data set is the the supervised training data set, containing pairs of maximum conductance samples $\mathbf{x}$ and the corresponding AP $\hat{V}_{\mathrm{m}}(t)$ that was obtained from the simulator. Sobol' sequences were used to generate 40,000 maximum conductance samples, containing 20,000 maximum conductance samples between 0% and 200% of the original values and 20,000 maximum conductance samples between 50% and 150% of the original values. The first covers a range that was considered plausible in terms of physiology and pathology (*Britton et al., 2017*; *Tomek et al., 2019*), and in terms of pharmacology (where full block is plausible). The latter covers a range that was considered particularly relevant in line with experimental calibration results presented in *Tomek et al., 2019* and patch clamp measurements of channels that were exposed to 30 clinical drugs blocks in up to the fourfold of the maximum free therapeutic concentration were analyzed in agreement with the CiPA paradigm (*Li et al., 2019*; *Crumb et al., 2016*). The `SALib-Sensitivity Analysis Library` (*Herman and Usher, 2017*) was used in the entire study for the generation of samples based on Sobol' sequences. For each maximum conductance sample, simulations were performed to obtain the corresponding 40,000 APs. APs with a transmembrane potential difference of more than 10% of the amplitude between $t = 0$ and 1000 ms (indicative of an AP that is far away from full repolarization) were excluded. This, however, applied to only 116 APs.

Starting from the original APs in data set #1, the data were first extended by 10 ms from $t \in [0, 1000]$ ms to $t \in [-10, 1000]$ ms to enable some extrapolation of $V_{\mathrm{m}}$ and hence a better alignment of the depolarization; for $t \in [-10, 0]$ ms the initial resting membrane potential $V_{\mathrm{m}}(0)$ was held constant. Then, the data were non-uniformly resampled from the original uniformly simulated APs, to emphasize the depolarization slope with a high accuracy while lowering the number of repolarization samples. For this purpose, we resamled the APs to 4 kHz for $t \in [-20, -5)$ ms (resting phase) and 10 kHz for $t \in [-5, 20)$ ms (depolarization phase) to 4 kHz. The repolarization phase ($t \in [20, 1000]$ ms) was also resampled to 4 kHz.

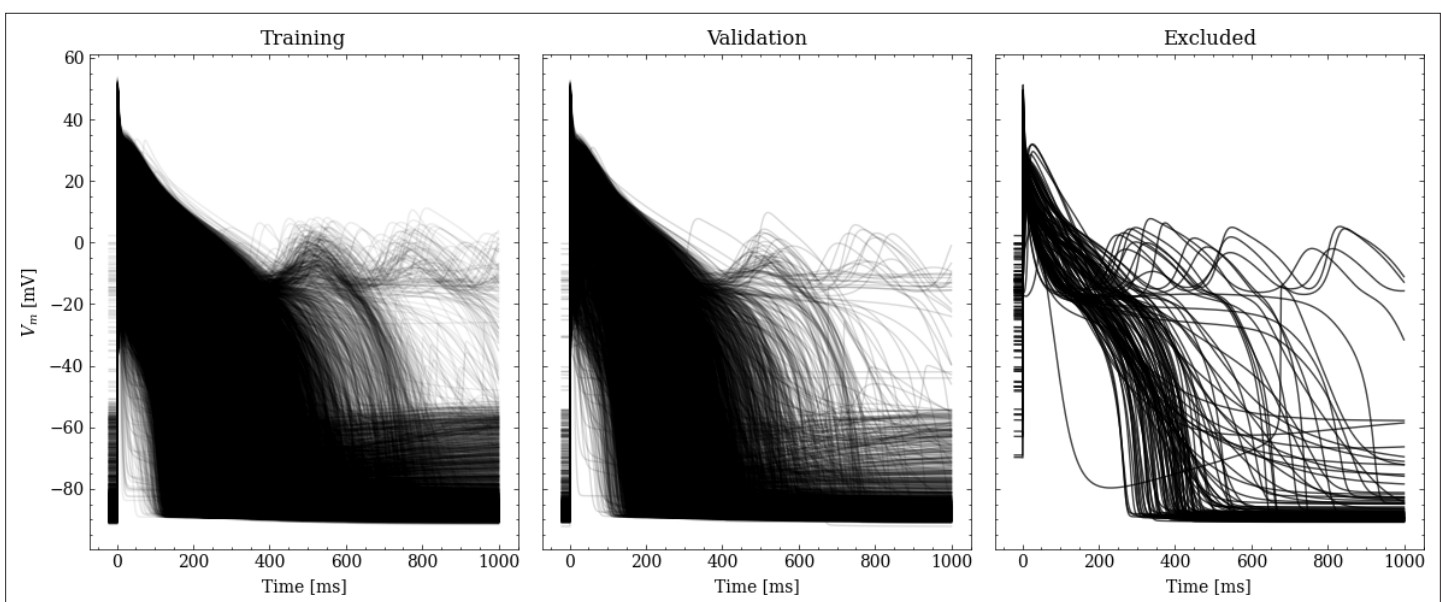

**Figure 2.** Processed APs used for training and validation (left and center). Additionally we show the excluded APs on the right (see text for description of the exclusion criteria).

**Table 1.** AP biomarkers and their experimental ranges used to generate the population of synthetic cardiomyocytes.

These were adopted from *Passini et al., 2017*. Experimental data were collected at 37°C in small right ventricular trabeculae and papillary tissue preparations obtained from healthy human hearts during pacing with a cycle length of $1000\ \mathrm{ms}$ (*Britton et al., 2017*; *O'Hara et al., 2011*).

| AP biomarker | Unit | Min | Max |
|---|---|---|---|
| *RMP* | mV | -95 | -80 |
| *dVmMax* | mVms$^{-1}$ | 100 | 1000 |
| *Peak* | mV | 10 | 55 |
| *APD$_{40}$* | ms | 85 | 320 |
| *APD$_{50}$* | ms | 110 | 350 |
| *APD$_{90}$* | ms | 180 | 440 |
| *Tri$_{90-40}$* | ms | 50 | 150 |

From the initial training data set, 20% were randomly chosen to be used for validation leaving 31908 pairs of maximum conductances and corresponding APs for training. *Figure 2* shows processed APs that were used for training, validation and the APs excluded due to missing full repolarization as described above.

## Synthetic test data (#2/#3)

Two sets of synthetic data were created using the simulator and each of the sets consisted of control and drug data with pairs of maximum conductances and corresponding APs.

The control data were the same in both sets. They were created using an experimentally calibrated population of 100 synthetic cardiomyocytes (*Britton et al., 2013*; *Muszkiewicz et al., 2016*; *Gemmell et al., 2016*). and to this end, Sobol' sequences were used to generate samples of maximum conductances with values between 50% and 150% of the original values. Maximum conductance samples that produced APs without abnormalities (checked for the last two consecutive APs; see AP biomarkers and abnormalities), and with seven biomarker values (derived from the last AP; see AP biomarkers and abnormalities) in agreement with experimental ranges (*Table 1*) were included in the population. Please note that the experimental ranges were not derived from the data set described in Experimental data (#4).

Data set #2: The motivation for creating data set #2 was to evaluate the emulator on data of normal APs. Drug data were created using 100 synthetic drugs represented by a set of pharmacological parameters. Each synthetic drug was built to have four different targets, with all channels, pumps, and exchangers related to the emulator inputs considered as potential targets. To this end, 100 samples of four pharmacological parameters, each with values between 0.5 (50% block) and 1.5 (50% enhancement) were randomly generated. The synthetic drugs were applied to the entire synthetic cardiomyocyte population by scaling each of the relevant control maximum conductances with the corresponding pharmacological parameter. The samples that produced APs without abnormalities (checked for the last two consecutive APs; see AP biomarkers and abnormalities) were included in the data set. No sample was excluded and thus, the data set consisted of 100 control data pairs and 10,000 drug data pairs.

Data set #3: The motivation for creating data set #3 was to test the emulator on data of abnormal APs showing the repolarization abnormality EAD. This is considered a particularly relevant AP

**Table 2.** Pharmacological parameter samples (synthetic drugs) with scaling factors for $G_{kr}$ and $P_{ca}$ to generate the drug data of data set #3.

| ID | 1 | 2 | 3 | 4 | 5 | 6 | 7 | 8 | 9 | 10 |
|---|---|---|---|---|---|---|---|---|---|---|
| $G_{kr}$ | 0.05 | 0.06 | 0.07 | 0.08 | 0.09 | 0.10 | 0.11 | 0.12 | 0.13 | 0.14 |
| $P_{ca}$ | 1.20 | 1.22 | 1.24 | 1.26 | 1.28 | 1.30 | 1.32 | 1.34 | 1.36 | 1.38 |

abnormality in pharmacological studies because of their role in the genesis of drug-induced ventricular arrhythmia's (*Weiss et al., 2010*). Drug data were created using 10 synthetic drugs with the hERG channel and the Cav1.2 channel as targets. To this end, 10 samples with pharmacological parameters for $G_{Kr}$ and $P_{Ca}$ (*Table 2*) were generated and the synthetic drugs were applied to the entire synthetic cardiomyocyte population by scaling $G_{Kr}$ and $P_{Ca}$ with the corresponding pharmacological parameter. Of the 1000 APs simulated, we discarded APs with a transmembrane potential difference of more than 10% of the amplitude between $t = 0$ and 1000 ms (checked for the last AP), indicative of an AP that is far away from fully repolarizing within 1000 ms. This left us with 950 APs, 171 of which exhibit EAD (see EAD classification).

## Experimental data (#4)

In the experimental data set, APs were recorded in small right ventricular trabeculae and papillary tissue preparations that were isolated from healthy human hearts (*Orvos, 2019*). The hearts were obtained from organ donors whose hearts were explanted to obtain pulmonary and aortic valves for transplant surgery. Before cardiac explantation, organ donors did not receive medication apart from dobutamine, furosemide, and plasma expanders. Proper consent was obtained for use of each individual's tissue for experimentation. The conventional microelectrode technique was used for AP recordings and all measurements were carried out at 37°C. Stimulation of APs was done at a pacing cycle length of 1000 ms using a pair of platinum electrodes that provided rectangular current pulses of 2 ms duration. To allow the preparations to equilibrate, stimuli were delivered for at least 60 min before the measurements started. Measurements were performed under control conditions and after administration of five channel-blocking drugs at one concentration in multiple preparations. Drugs were cisapride (30 nM), dofetilide (10 nM), sotalol (30 $\mu$m), terfenadine (1 $\mu$m), and verapamil (300 nM). The last 10 consecutive APs of each measurement were analyzed to quantify the beat-to-beat variability. Overall, the beat-to-beat variability was found to be small (standard deviation in all APs below 7 mV before the peak due to time alignment mismatch and below 2 mV after the peak) and thus, the last 10 consecutive APs of each measurement were averaged. In most of the preparations, the standard deviation between beats did not vary over time and thus, no temporal correlation of noise was assumed. Averaging also reduced the noise level. The experimental data set contained one pair of averaged control and drugged AP per preparation per drug. Pairs were excluded if the biomarker values (see AP biomarkers and abnormalities) of the control or the drugged AP were not in the range found in the training data (see Training). This applied to seven pairs and the final data set contained three pairs for cisapride, dofetilide, sotalol, and terfenadine, and one pair for verapamil. All measurements were performed at the University of Szeged, Hungary, and conformed to the principles of the Declaration of Helsinki. Experimental protocols were approved by the University of Szeged and by the National Scientific and Research Ethical Review Boards (No. 51-57/1997 OEj and 4991-0/2010-1018EKU [339/PI/010]).

An overview of all utilized data is given in *Table 3*.

**Table 3.** Summary of the data used in this study, along with their usage and the number of valid samples.

Note that each AP is counted individually, also in cases of control/drug pairs.

| ID | Description | Usage | Origin | Samples |
|---|---|---|---|---|
| #1 | Training/validation data | Training and validating the emulator, choosing the best architecture (Architecture) | Simulation | 39,884 |
| #2 | Synthetic drug data, normal APs | Testing forward and inverse performance for normal APs ('Forward problem' and 'Inverse problem based on synthetic data') | Simulation | $10^4$ |
| #3 | Synthetic drug data, including EAD APs | Testing forward performance of abnormal (EAD) APs ('Forward problem') | Simulation | 950 |
| #4 | Experimental cardiomyocytes | Testing and comparing the inverse performance with data published by the CiPA initiative (*Li et al., 2017*; *Chang et al., 2017*; 'Inverse problem based on experimental data') | *Orvos, 2019* | 26 |

## Architecture

The emulator takes maximum conductances of channels, pumps, and exchangers as inputs and computes the corresponding AP ($V_\mathrm{m}(t)$) after the last stimulus as output. It was trained to represent human ventricular cardiomyocytes under control and drugged conditions and the inputs were selected based on two assumptions: (1) The kinetics of channels are preserved, while the number of channels, pumps, and exchangers vary due to different expression levels (*Syed et al., 2005*; *Groenendaal et al., 2015*; *Krogh-Madsen et al., 2016*). These numbers are captured in the simulator by the maximum conductance parameters (or permeability parameters but maximum conductance is used as general term here for the sake of simplicity) that determine the respective current densities; (2) Channels, pumps, and exchangers are potential (intended and unintended) drug targets and the interaction between drugs and their targets can be described by a scaling of the related maximum conductances (*Brennan et al., 2009*). The corresponding scaling factors are pharmacological parameters. These assumptions allowed us to focus on maximum conductances and we considered those as inputs which either the AP is sensitive to ($G_\mathrm{Na}$, $G_\mathrm{NaL}$, $P_\mathrm{Ca}$, $G_\mathrm{to}$, $G_\mathrm{Kr}$, $G_\mathrm{K1}$, $G_\mathrm{NCX}$, $P_\mathrm{NaK}$) or which are related to common drug targets ($G_\mathrm{Na}$, $G_\mathrm{NaL}$, $P_\mathrm{Ca}$, $G_\mathrm{to}$, $G_\mathrm{Kr}$, $G_\mathrm{Ks}$, $G_\mathrm{K1}$; *Crumb et al., 2016*) leading to the following selection: $G_\mathrm{Na}$, $G_\mathrm{NaL}$, $P_\mathrm{Ca}$, $G_\mathrm{to}$, $G_\mathrm{Kr}$, $G_\mathrm{Ks}$, $G_\mathrm{K1}$, $G_\mathrm{NCX}$, and $P_\mathrm{NaK}$. AP sensitivity was quantified using a global sensitivity analysis (GSA; see.Appendix 2) and the inclusion threshold was a total-effect Sobol' sensitivity index (ST) above 0.1 with respect to any of the considered biomarkers (see.Appendix 1).

Several emulator architectures were tried on the training and validation data sets and the final choice was hand-picked as a good trade-off between high accuracy on the validation set (#1) and low computational runtime cost. We decided to utilize a two-stage emulator architecture: First, the maximum conductances $\mathbf{x}$ – normalized to the range $\mathbf{x}_i \in [-0.5,\ 0.5]$ – are encoded using a first NN ($\Theta_1$) into a latent representation $\vartheta$. Second, this intermediate representation parameterizes a function $f_\vartheta : \mathbb{R} \to \mathbb{R}$ defined by a second NN ($\Theta_2$) that can be continuously evaluated to receive the emulated AP at time $t$. To help the second NN in computing the fast depolarization, a simple depolarization term (tanh) is added to $f_\vartheta$. The parameters of this depolarization function are slope ($d_1$), offset ($d_2$), and amplitude ($d_3$), and are created by encoding the parameters through the first network, similar to the latent code. The AP approximated by the emulator is thus defined by

$$V_\mathrm{m}(t) := f_\vartheta(t) + \tanh(d_1^2(t - d_2)) + d_3^2. \tag{1}$$

A schematic drawing of the emulator architecture is provided in 2. Splitting the network into two parts — one for encoding the parameters into a latent space and a second one for evaluating $f_\vartheta$ — allowed us to give the emulator enough complexity without markedly increasing the computational cost: in

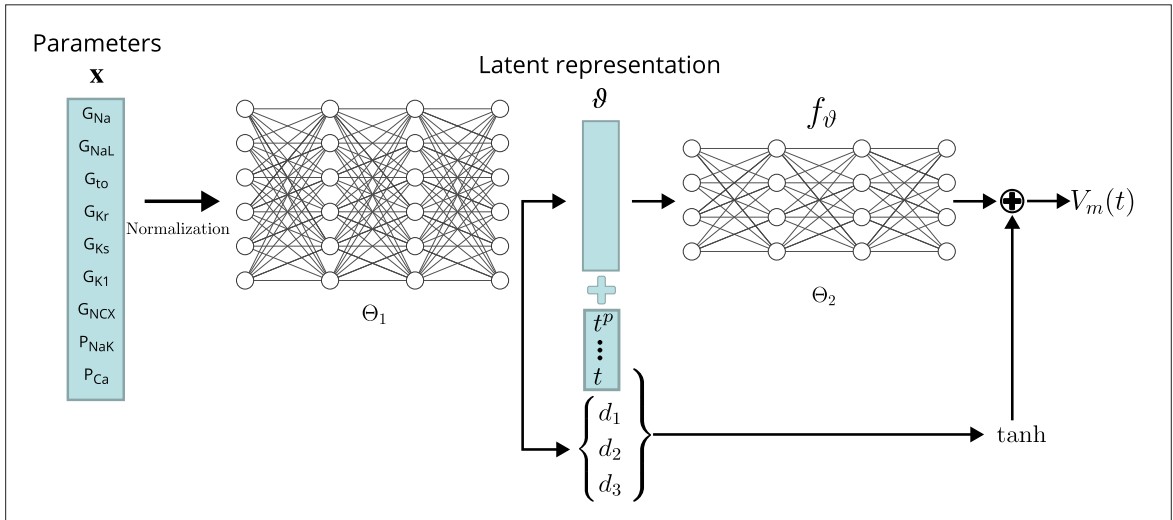

**Figure 3.** Conceptual architecture of the neural network emulator. The maximum conductances $\mathbf{x}$ are encoded into depolarization parameters $d_i$ and a latent space representation $\vartheta$ that uniquely defines the time series functional $f_\vartheta : \mathbb{R} \to \mathbb{R}$. The time is normalized and encoded in polynomials up to degree 8 ($t^p$ for $p \in \{1, 2, \ldots 8\}$), before being appended to the latent code $\vartheta$. $f_\vartheta$ is then used in conjunction with the depolarization helper **tanh** to approximate the AP $V_\mathrm{m}(t)$.

most cases, it is desirable to compute the whole AP in the entire range, for example [0, 1000] ms, and not only at a single time step. Creating a single network that computes the mapping from maximum conductances to the transmembrane potential at a single time step (compare *Figure 3*) is either orders of magnitudes slower than encoding the parameters into a latent vector (only done once per AP) or would require to reduce the complexity of the network, which led to inaccurate emulations in the validation. The additional depolarization term was introduced to address the difficulty of fitting the depolarization phase during training and decreased the required training time substantially. Note that the mapping from maximum conductances $\mathbf{x}$ to depolarization parameters $\{d_1, d_2, d_3\}$ is also learned through $\Theta_1$.

The exact architecture employed – chosen by a cross-validation approach (see Validation) – comprised a first network ($\Theta_1$) of four fully connected layers of 256 neurons, each to encode the parameters into the latent vector $\vartheta \in \mathbb{R}^{256}$. This first network additionally generates the parameterization for the depolarization model $\{d_1, d_2, d_3\}$. The second network, computing the APs from the latent representation ($\Theta_2$), consisted of four fully connected layers of 64 neurons each. Exponential linear unit (ELU) activation functions *Clevert et al., 2016* were used for all layers, except for the final non-linear layer, which was modeled using a $\tanh$ activation function followed by a $[-1, 1]^{64} \to \mathbb{R}$ linear layer.

## Training

Although different regularization schemes such as variational losses (e.g. $\frac{dV_m}{dt}$) were tried, the wealth of training data allowed us to define the training loss purely in terms of mean-squared-error (MSE)

$$\mathcal{L}(\Theta_1, \Theta_2) = \frac{1}{2|\mathbb{T}|} \sum_{(\mathbf{x}, \hat{V}_m) \in \bar{X} \subset X} \sum_{t \in \mathbb{T}} \left( f_{\vartheta_{\Theta_1}(\mathbf{x}), \Theta_2}(t) - \hat{V}_m(t) \right)^2 dt, \tag{2}$$

where $f_{\vartheta_{\Theta_1}(\mathbf{x}), \Theta_2}$ describes the output of the emulator using the current NN weights $\Theta_1/\Theta_2$, $\bar{X}$ refers to the current training batch and $X$ is the training data set containing both target APs $\hat{V}_m$ and corresponding maximum conductances $\mathbf{x}$. For the training, increasing batch sizes ($|\bar{X}|$ 1250 to 1800) were used, both in terms of the entire AP and considering only subsets over time $T \subset \mathbb{T}$ from $\frac{1}{16}|\mathbb{T}|$ to $|\mathbb{T}|$, resulting in a reduction of training time needed. The neural network was trained for a total number of 5000 epochs using the first-order gradient-based algorithm ADAM *Kingma and Ba, 2017*. The training time was approximately 4 h on the GPU specified in Computational performance.

## Validation

A cross-validation approach was used to quantify and compare the performance of various emulator architectures. The validation was based on 20% of the initial training data set (7976 pairs of maximum conductances and corresponding APs, see Training). For each pair, the emulated AP $V_m$ was compared against the simulated AP $\hat{V}_m$, given the same maximum conductances. The mismatch was quantified by the root-mean-squared error (RMSE) defined as

$$\text{RMSE}(V_m, \tilde{V}_m) := \sqrt{\frac{1}{|T|} \int_T \left( V_m(t) - \hat{V}_m(t) \right)^2 dt}.$$

The mismatch was also quantified in terms of AP biomarkers $\mathbf{b} \in \mathbb{R}^N$ (see AP biomarkers and abnormalities) and normalized maximum conductances $\mathbf{x}$ (see Time series fitting and estimation of maximum conductances and pharmacological parameters) in which case the RMSE was defined as

$$\text{RMSE}(\mathbf{b}, \tilde{\mathbf{b}}) := \sqrt{\frac{1}{N} \|\mathbf{b} - \tilde{\mathbf{b}}\|_2^2} \quad \text{and} \quad \text{RMSE}(\mathbf{x}, \tilde{\mathbf{x}}) := \sqrt{\frac{1}{N} \|\mathbf{x} - \tilde{\mathbf{x}}\|_2^2},$$

for $N$ samples.

## Time series fitting and estimation of maximum conductances and pharmacological parameters

Time series fitting is the basis for solving the inverse problem. To fit a given AP $\hat{V}_m$, defined on a subset of the trained domain $\hat{\mathbb{T}} \subset \mathbb{T}$, the first step was to choose a trial set of maximum conductances $\mathbf{x}_0$.

Then, for the given trial set, the corresponding AP was emulated and the trial set of maximum conductances was iteratively updated by solving the following minimization problem:

$$\min_{\mathbf{x},t_0} \frac{1}{2|\hat{\mathbb{T}}|} \sum_{t\in\hat{\mathbb{T}}} \left(V_m(\mathbf{x}, t-t_0) - \hat{V}_m(t)\right)^2 dt + \frac{\lambda_{\mathbf{x}_0}}{2}\|\mathbf{x}-\hat{\mathbf{x}}\|_2^2 + \delta_{[-0.5,0.5]}(\mathbf{x}),$$ (3)

where $V_m(\mathbf{x},t) := f_{\vartheta_{\Theta_1}(\mathbf{x}),\Theta_2}(t)$ is a shorthand for the emulator approximation function and $t_0$ is a temporal offset parameter helping in fitting the exact depolarization timing. Here, $\delta_{[-0.5,0.5]}(\mathbf{x})$ is the element-wise indicator function on the normalized feasible parameter space $[-0.5, 0.5]$. The minimization was done using ADAM (*Kingma and Ba, 2017*) combined with a projection on the feasible space.

To estimate maximum conductances for a given control AP, the control AP was fitted using the original maximum conductance values as initial guesses and priors: $\mathbf{x}_0 = \hat{\mathbf{x}} = \mathbf{0}$. To estimate maximum conductances for a given drugged AP, the drugged AP was fitted using the maximum conductances estimated for the respective control AP as initial guesses and priors: $\mathbf{x}_0 = \hat{\mathbf{x}} = \mathbf{x}^c$. The pharmacological parameters (scaling factors of control maximum conductances) were computed as element-wise ratios between the drugged and control maximum conductances $(\mathbf{s})_i := \frac{(\mathbf{G}^d)_i}{(\mathbf{G}^c)_i}$ but here, $\mathbf{G}^d$ and $\mathbf{G}^c$ are the non-normalized maximum conductances, where $(\mathbf{G}^c)_i > 0$.

Since multiple sets of maximum conductances produced similarly good fits of the given AP, the parameter $\lambda_{\mathbf{x}_0}$ was introduced which minimizes the difference between original and control maximum conductances and between control and drugged maximum conductances, respectively. The value was

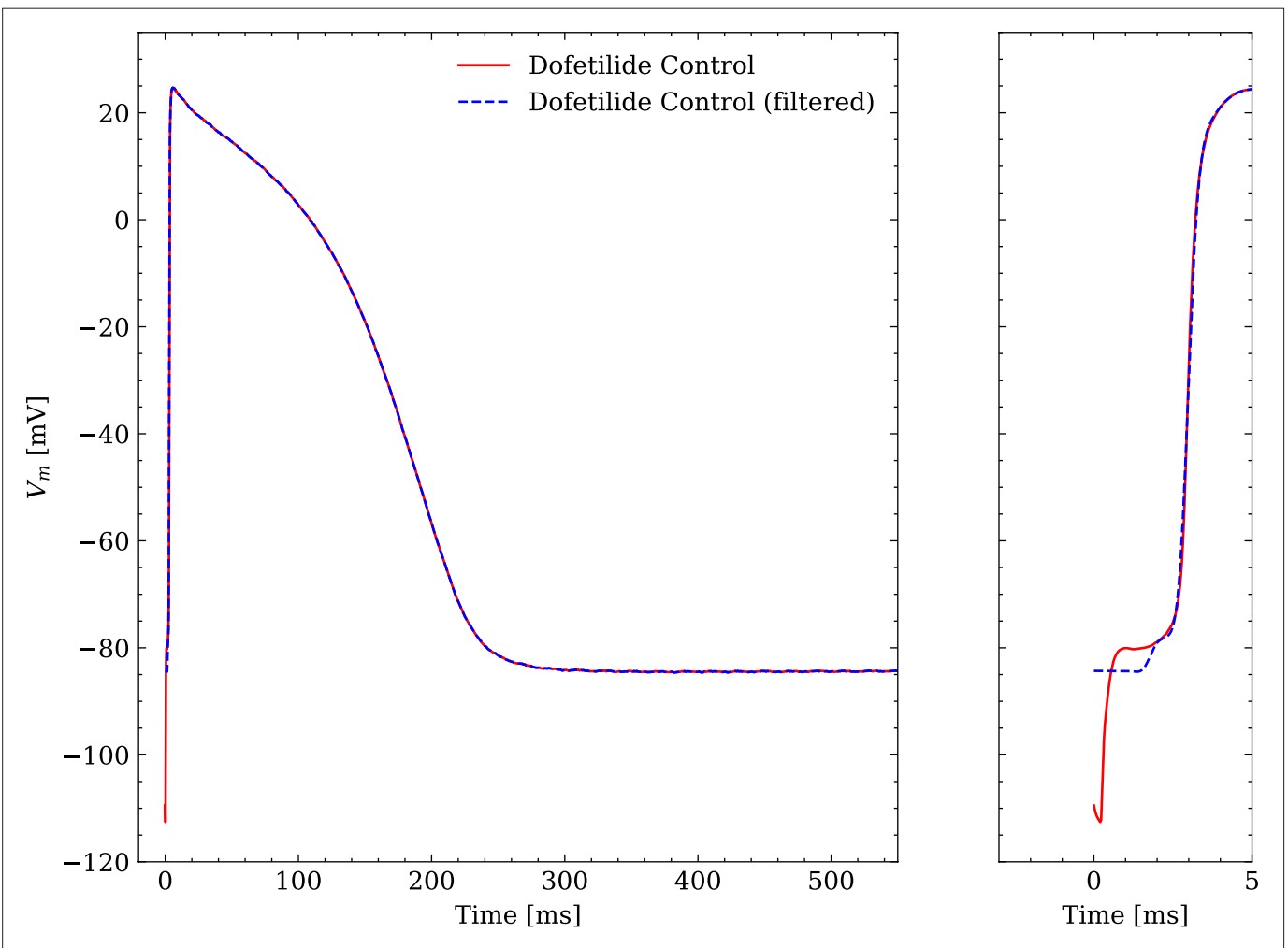

**Figure 4.** Comparison of an averaged raw and an averaged filtered experimental AP. One dofetilide control AP is shown as example.

chosen to be $\lambda_{\mathbf{x}_0} = 10$ with respect to the highest accuracy found for the synthetic data set #2 that was generated for the evaluation (see Synthetic test data (#2/#3)).

## Evaluation

The evaluation was performed for forward and inverse problems in pharmacological studies on synthetic and experimental data.

The raw experimental data were obtained without filtering but some filtering was applied before interfacing the data with the emulator. The APs contained a stimulus artifact between 0 and 1.5 ms that was filtered as follows. For each of the last 10 consecutive APs, the transmembrane potential $V_m$ closest to the end of the recorded time series was defined as the resting transmembrane potential $RMP$ and $V_m(t) = RMP$ was set for $t \in [0, 1.75]$ms. Then, the APs were resampled at 100 kHz and a low-pass filtering was performed with a second-order butterworth filter (cutoff at 2.5 kHz) to reduce the high-frequency noise of the signal. Finally, the filtered APs were averaged and the averaged AP was again resampled at 1 kHz for $t \in [15, 1000]$ms (repolarization) and 100 kHz in $t \in [0, 15]$ms (depolarization). An example comparison of a raw and a filtered averaged AP is given in *Figure 4*.

### Computational performance

The simulation of a single AP (see Simulator) sampled at a resolution of 20 kH$_z$ took 293 s on one core of a AMD Ryzen Threadripper 2990 WX (clock rate: 3.0 GH$_z$) in CARPentry. Adaptive timestep solver of variable order, such as implemented in `Myokit` (*Clerx et al., 2016*), can significantly lower the simulation time (30 s for our setup) by using small step sizes close to the depolarization (phase 0) and increasing the time step in all other phases. The emulation of a steady state AP sampled at a resolution of 20 KH$_z$ for $t \in [-10, 1000]$ ms took 18.7 ms on a AMD Ryzen 7 3800 X (clock rate: 3.9 GH$_z$) and 1.2 ms on a Nvidia A100 (Nvidia Corporation, USA), including synchronization and data copy overhead between CPU and GPU.

The amount of required beats to reach the steady state of the cell in the simulator has a major impact on the runtime and is not known a-priori. On the other hand, both simulator and emulator runtime linearly depends on the time resolution, but since the output of the emulator is learned, the time resolution can be chosen at arbitrarily without affecting the AP at the sampled times. This makes direct performance comparisons between the two methodologies difficult. To still be able to quantify the speed-up, we ran `Myokit` using 100 beats to reach steady state, taking 3.2 s of simulation time. In this scenario, we witnessed a speed-up of 171 and $2 \cdot 10^3$ of our emulator on CPU and GPU, respectively (again including synchronization and data copy overhead between CPU and GPU in the latter case). Note that both methods are similarly expected to have a linear parallelization speedup across multiple cells.

For the inverse problem, we parallelized the problem for multiple cells and keep the problem on the GPU to minimize the overhead, achieving emulations (including backpropagation) that run in 120 s per AP at an average temporal resolution of 2 KH$_z$. We consider this the peak performance which will be necessary for the inverse problem in Inverse problem based on synthetic data.

### Forward problem

The emulator evaluation for the forward problem, i.e. to find the drugged AP for given pharmacological parameters, was only performed on synthetic data since maximum conductances were not available experimentally. The maximum conductances of data sets #2 and #3 were used to consider data with normal APs and with abnormal APs exhibiting EADs. Pharmacological parameters are not inputs of the emulator but drugged maximum conductances that were computed as control maximum conductances scaled by the given pharmacological parameters (see Synthetic test data (#2/#3)). These were used to emulate drugged APs. The RMSE was used to quantify the mismatch between the emulated and the ground truth AP.

### Inverse problem

The emulator evaluation for the inverse problem, that is to find the pharmacological parameters for given control and drugged APs (through optimization), was performed on both synthetic and experimental data. When using synthetic data, the data set #2 was used including data with normal APs. First, control and drugged maximum conductances were estimated based on control and drugged

APs and then, pharmacological parameters were computed as ratios of drugged and control conductances (see Time series fitting and estimation of maximum conductances and pharmacological parameters). The mismatch between estimated and ground truth maximum conductances were quantified using the error that is defined through

$$\mathbf{e}_n := \left( \mathbf{x} - \hat{\mathbf{x}} \right),\qquad(4)$$

where $\mathbf{x}$ and $\hat{\mathbf{x}}$ are the normalized estimated and ground truth maximum conductances (see Architecture). Similarly, we computed the mismatch between estimated and ground truth scaling factor vectors ($\mathbf{s}$ and $\hat{\mathbf{s}}$ respectively) as

$$\mathbf{e}_s := \left( \mathbf{s} - \hat{\mathbf{s}} \right).\qquad(5)$$

When using experimental data, maximum conductances and pharmacological parameters were estimated in the same way but due to a lack of experimental maximum conductances, the mismatch between estimated and ground truth values could not be quantified. Instead, the estimated pharmacological parameters were compared with distributions computed from data published within the CiPA initiative (*Li et al., 2017*; *Chang et al., 2017*) (CiPA distributions). The data set (https://github.com/FDA/CiPA/tree/master/Hill_fitting/results; *Chang and Li, 2017*) includes 2000 $IC_{50}$ values and Hill coefficients for each drug and for up to seven targets ($I_{Na}$, $I_{NaL}$, $I_{CaL}$, $I_{to}$, $I_{Kr}$, $I_{Ks}$, $I_{K1}$). The pore-block model (*Brennan et al., 2009*) was used to obtain the corresponding scaling factors.

## Results
### Evaluation
#### Forward problem
The emulator evaluation for the forward problem was only done on synthetic data and both data sets #2 and #3 (see Synthetic test data (#2/#3)) were used to analyze the solution accuracy for normal and abnormal APs exhibiting EADs.

The data set #2 was used first and *Figure 5* illustrates the distribution of RMSEs between emulated and ground truth drugged APs. In total $10^4$ APs were emulated in 0.6 s. The average RMSE over all APs was 0.47 mV and only for a few APs the RMSE was >1 mV with 1.5 mV being the maximum. Largest mismatches were located in the phases 0 and 3 of the AP. While the mismatches in phase 3 were simply a result of imperfect emulation, the mismatches in phase 0 were a result of the difficulty in matching the depolarization time exactly.

*Figure 6* shows the distribution of biomarker mismatches between emulated and ground truth drugged APs. The low RMSEs between the APs translated into low RMSEs between the AP biomarkers.

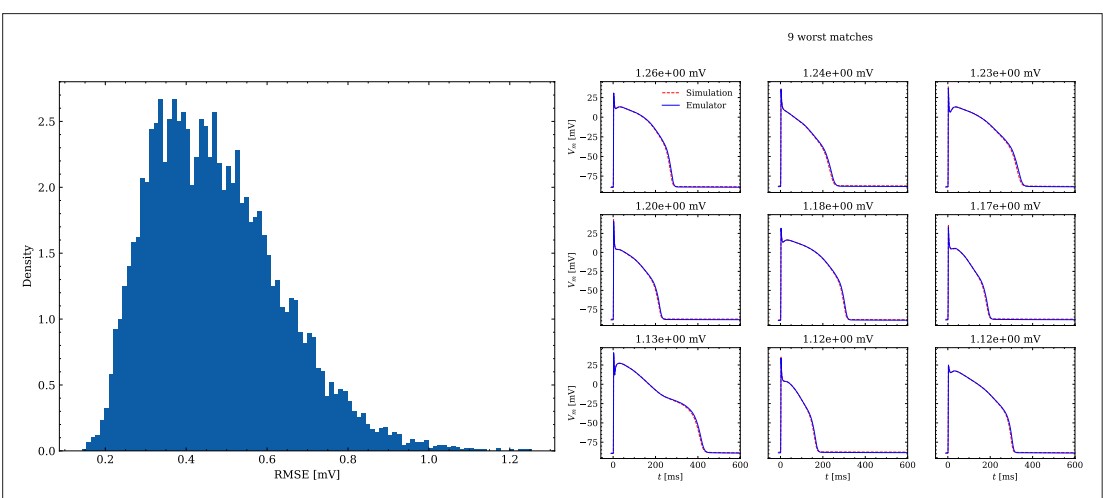

**Figure 5.** Analysis of solution accuracy of the forward problem on synthetic data including normal APs (drug data of data set #2). Left: histogram of RMSEs for the APs, right: APs with the largest RMSEs. The RMSE is given above each subplot.

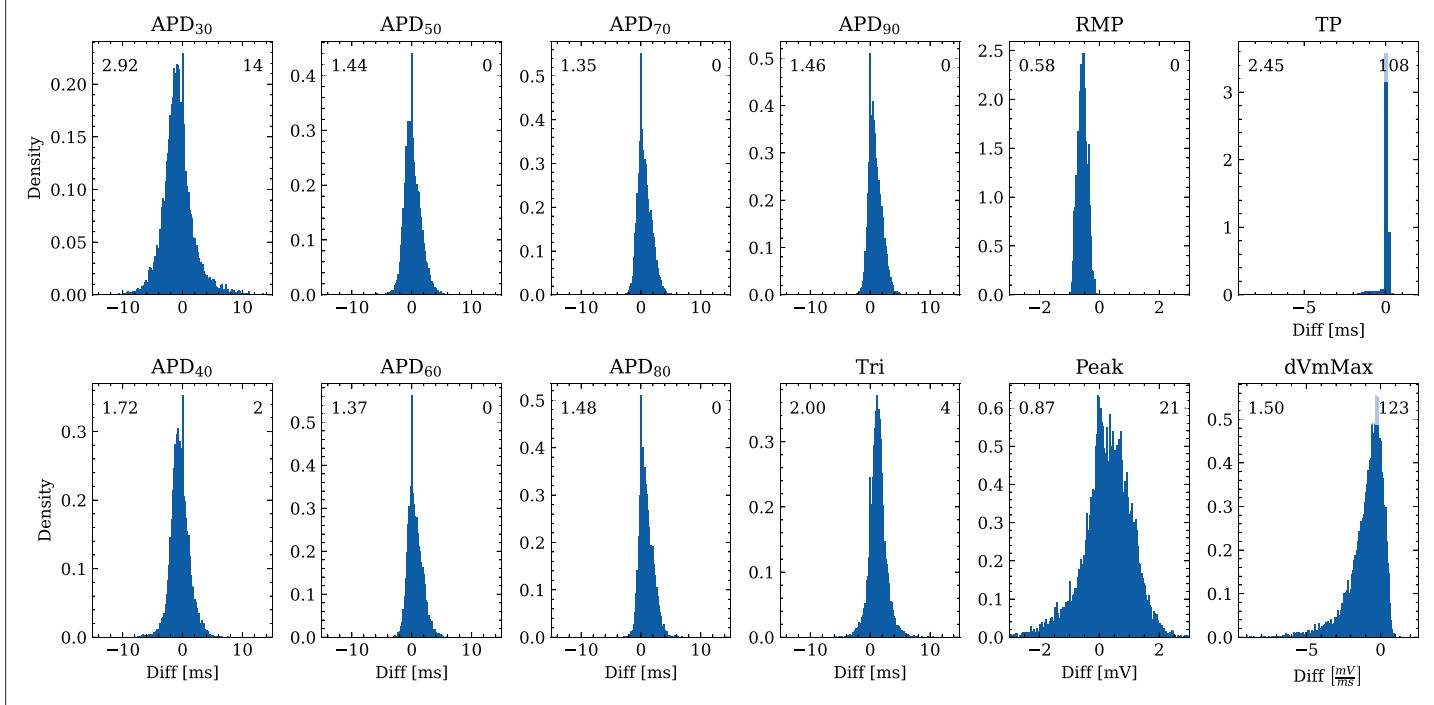

**Figure 6.** Analysis of solution accuracy of the forward problem on synthetic data including normal APs (drug data of data set #2) with respect to AP biomarkers. Histograms of mismatches for each biomarker are shown and the RMSE is given in the upper left corner. The number in the right upper corner denotes the number of outliers of the 10,000 samples which lie outside the shown ranges.

Likewise, the difficulty in exactly matching the depolarization time leads to elevated errors and more outliers in the biomarkers influenced by the depolarization phase (*TP* and *dVmMax*).

The data set #3 was used second and Appendix 3 shows all emulated APs, both containing the EAD and non-EAD cases. The emulation of all 950 APs took 0.76 s on the GPU specified in Training We show the emulation of all maximum conductances and the classification of the emulation. The comparison with the actual EAD classification (based on the criterion outlined in Appendix 1) results in true-positive (EAD both in the simulation and emulation), false-negative (EAD in the simulation, but not in the emulation), false-positive (EAD in the emulation, but not in the simulation) and true-negative

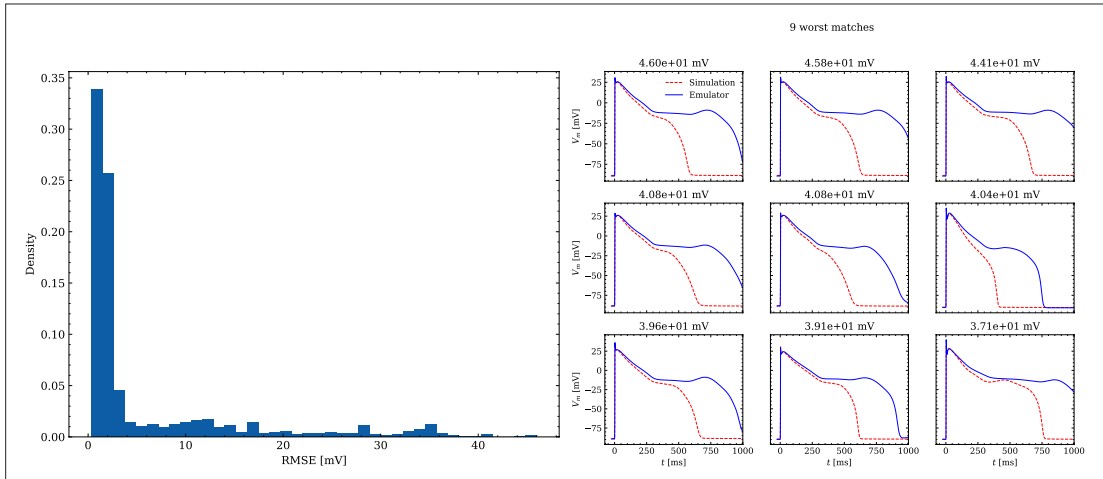

**Figure 7.** Analysis of solution accuracy of the forward problem on synthetic data including abnormal APs exhibiting EADs (subset of data set #3). Left: histogram of RMSEs for the APs, right: APs with the largest RMSEs. Of the 171 emulated APs, 124 exhibit the expected EADs (based on the criterion outlined in Appendix 1). The RMSE is given above each subplot. All emulated APs are shown in Appendix 3.

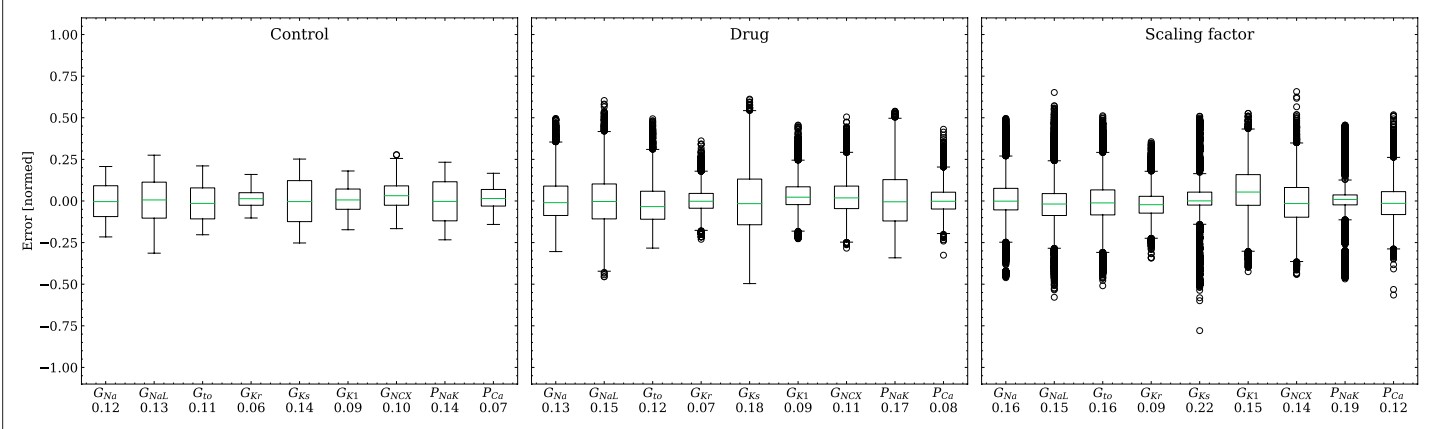

**Figure 8.** Analysis of solution accuracy of the inverse problem on synthetic data (data set #2). Left: boxplot of errors between normalized estimated and ground truth control maximum conductances, middle: boxplot of errors between normalized estimated and ground truth drugged maximum conductances, right: boxplot of errors between estimated and ground truth pharmacological parameters. Error definitions are given in (4) and (5). The RMSE over all data is given below each parameter.

(no EAD both in the emulation and simulation). The emulations achieved 72.5% sensitivity (EAD cases correctly classified) and 94.9% specificity (non-EAD cases correctly classified), with an overall acurracy of 90.8% (total samples correctly classified). A substantial amount of wrongly classified APs showcase a notable proximity to the threshold of manifesting EADs. *Figure 7* illustrates the distribution of RMSEs in the EAD APs between emulated and ground truth drugged APs. The average RMSE over all EAD APs was 14.5 mV with 37.1 mV being the maximum. Largest mismatches were located in phase 3 of the AP, in particular in emulated APs that did not fully repolarize.

## Inverse problem based on synthetic data

The emulator evaluation for the inverse problem was first done using synthetic data (data set #2, see Synthetic test data (#2/#3)). For this, we minimized (3) by using ADAM (*Kingma and Ba, 2017*) with no batching for $10^4$ iterations for all 100 cardiomyocytes times 100 drugs (i.e. $10^4$ APs), resulting in a total of $10^8$ emulations, taking approximately 3.5 h on the GPU specified in Training. Control and drugged APs could be fitted with an average RMSE of 0.8 mV. Largest mismatches were located in phase 0 and 3 of the AP for the reasons given above (see Forward problem). *Figure 8* shows the distribution of the errors between the estimated and the ground truth maximum conductances and pharmacological parameters. For both the maximum conductances (RMSE $\leq 0.18$) and the related pharmacological parameters (RMSE $\leq 0.22$), the errors were closely distributed around zero. However, the RMSEs increased from the control maximum conductance over the drugged maximum conductance to the pharmacological parameters and there were distinctive differences among maximum conductances and related pharmacological parameters with the smallest for $G_{Kr}$ and the largest for $G_{Ks}$.

## Inverse problem based on experimental data

The emulator evaluation for the inverse problem was then done on experimental data (Experimental data (#4)). Similar to the synthetic inverse problem, we optimized (3), this time for $5 \cdot 10^4$ and $2.5 \cdot 10^4$ epochs for control and drug APs for each drug sample. The total computational time spent on this task was 2.8 hr on the GPU specified in Training.

*Figure 9* shows the fitted and the ground truth APs for all drugs. Control and drugged APs could be fitted with average RMSEs shown in *Table 4*. The largest mismatch was located in phase 0 for most APs, which was the result from an imperfect matching of the exact depolarization timing.

*Figure 10* compares the estimated pharmacological parameters and the CiPA distributions. Pharmacological parameters that fell into the range spanned by $\mu \pm (0.15 + \sigma)$ of the CiPA distribution, where $\mu, \sigma$ are the distribution's mean and standard deviation respectively, were classified as successfully estimated while the others were classified as unsuccessfully estimated. In total, 50% of the pharmacological parameters could be estimated succesfully and while all pharmacological parameters

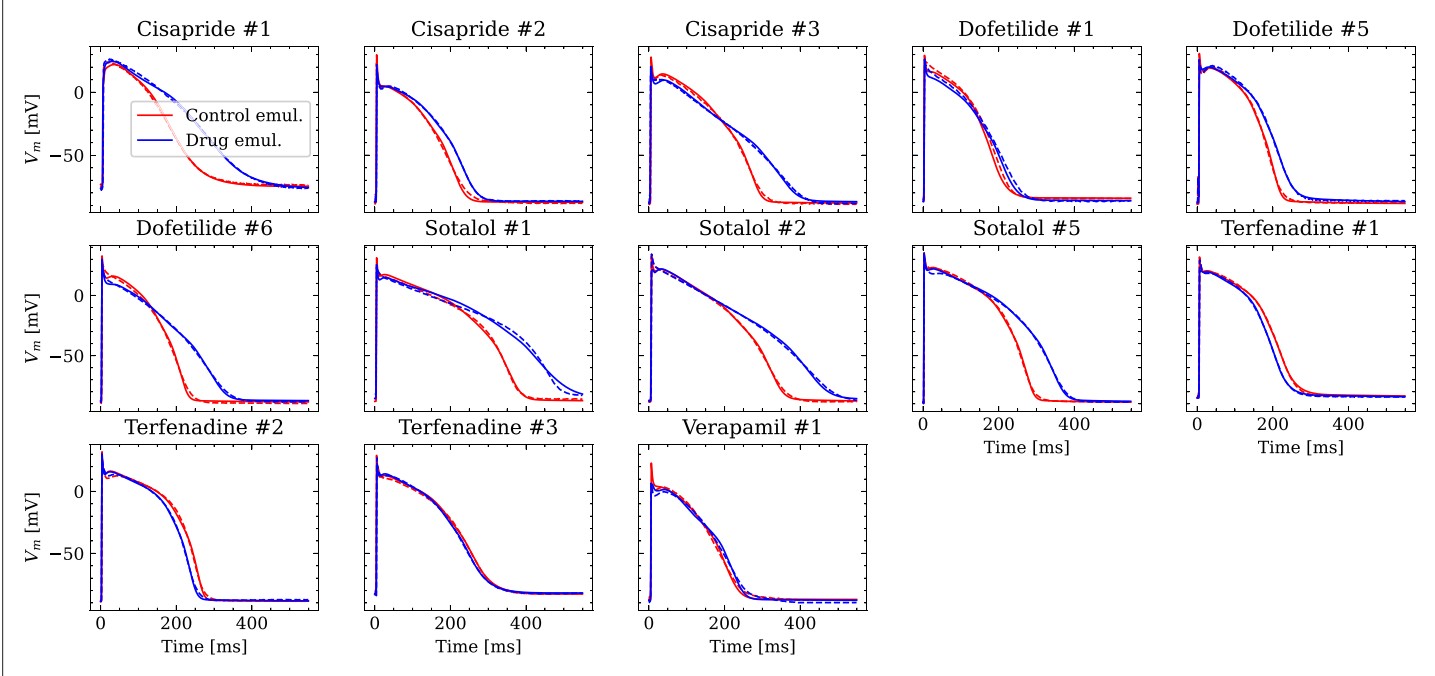

**Figure 9.** Analysis of fit quality of the inverse problem on experimental data. Comparison of the fitted APs (solid lines) and the experimental APs (dashed lines) at control (red) and after drug administration (blue) for all drugs.

related to $G_{Ks}$ could be successfully estimated, unsuccessfully estimated parameters were found across all maximum conductances, in particular related to $G_{K1}$ for which all pharmacological parameters could not be successfully estimated (*Table 5*).

## Discussion

NN emulation of the human ventricular cardiomyocyte AP was introduced and the applicability in pharmacological studies was investigated.

### Evaluation

The evaluated NN emulator showed highly accurate AP emulations for the forward problem on synthetic data. High accuracy was found in normal APs in data set #2 (average RMSE was 0.47 mV; *Figure 5*) and to a lesser extent also in abnormal APs exhibiting EADs: of the emulated EAD APs, 72.5% exhibited alignment with the abnormality, and the substantial majority of the remaining APs demonstrated pronounced proximity, while the average RMSE among the EAD APs was 14.5 mV (*Figure 7*). In comparison, the normal APs exhibiting no EAD in data set #3 could be reconstructed with an average RMSE of 4.02 mV and the overall accuracy of classifying EAD on emulated APs was 90.8%. Increasing the amount of training data within the relevant range could lead to further enhancements in accuracy for abnormal APs. Nevertheless, this observation demonstrates that the

**Table 4.** Average RMSE over control and drugged APs measured in all preparations per drug. All values in mV.

| Drug | RMSE control | RMSE drug |
|---|---|---|
| Cisapride | 1.53 | 2 |
| Dofetilide | 2.05 | 1.73 |
| Sotalol | 1.4 | 2.51 |
| Terfenadine | 1.22 | 1.08 |
| Verapamil | 1.93 | 2.21 |

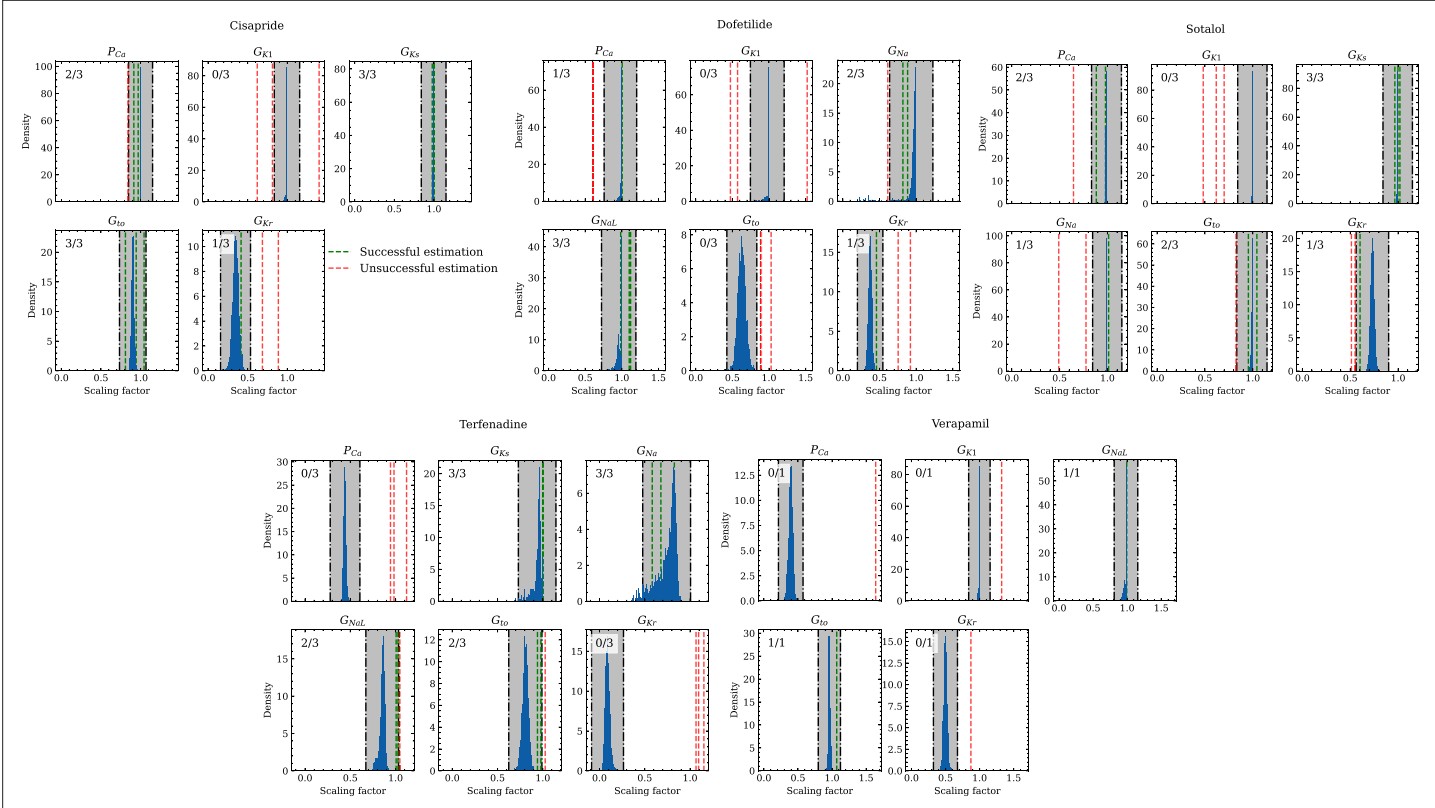

**Figure 10.** Analysis of solution accuracy of the inverse problem using experimental data. The histograms compare the estimated pharmacological parameters (dashed vertical lines) from data of multiple preparations with the CiPA distributions (blue; see Inverse problem). The black dash dotted vertical lines are the borders of the range (grey) that was used to determine if the estimation of the given pharmacological parameter was successful. The range is spanned up by $\mu \pm (0.15 + \sigma)$ of the CiPA distribution. Successfully estimated parameters are shown as green lines and unsuccessfully estimated parameters are shown as red lines. The number in the upper left corner indicates for how many preparations the parameters could be successfully estimated in relation to the total number of preparations for the given drug.

emulator is also capable of accounting for discontinuities of the response surface. This is particularly useful in pharmacological studies and a key advantage over existing emulation approaches (*Chang et al., 2015*; *Johnstone et al., 2016*; *Coveney and Clayton, 2018*; *Ghosh, 2018*; *Rasmussen, 2019*; *Coveney et al., 2021*).

The emulator was further evaluated for the inverse problem on synthetic and also on experimental data. Maximum conductances and related pharmacological parameters could be widely estimated with high accuracy on synthetic data (RMSE $\leq 0.18$ and $\leq 0.21$ for all maximum conductances and pharmacological parameters, respectively; *Figure 8*).

The RMSEs increased from the control maximum conductance over the drugged maximum conductance to the pharmacological parameters which may be because the estimation of drugged

**Table 5.** Pharmacological parameters related to maximum conductances that were considered successfullyor unsuccessfully estimated across all preparations and drugs.

For each channel, the drugs are stated forwhich respective data from the CiPA initiative were available. C, D, S, T, V, A mark cisapride, dofetilide, sotalol,terfenadine, verapamil, all drugs respectively.

| | Gna | GNaL | Gto | GKr | GKs | GK | PCa | Total |
|---|---|---|---|---|---|---|---|---|
| Successful | 6 | 6 | 8 | 3 | 9 | 0 | 5 | 37 |
| Unsuccessful | 3 | 1 | 5 | 10 | 0 | 10 | 8 | 37 |
| Ratio | 0.67 | 0.86 | 0.62 | 0.23 | 1 | 0 | 0.38 | 0.5 |

maximum conductances depends on the control maximum conductances and the computation of pharmacological parameters depends on both control and drugged maximum conductances (see Time series fitting and estimation of maximum conductances and pharmacological parameters) allowing that errors can propagate and amplify. Distinctive differences were observed among the maximum conductances and related pharmacological parameters and the largest RMSEs were found for $G_{Ks}$ throughout. This can be attributed to various degrees of parameter identifiability *Sarkar and Sobie, 2010*; *Zaniboni et al., 2010*; *Groenendaal et al., 2015*; *Jaeger et al., 2019*; *Whittaker et al., 2020* and the results agree with the GSA that indicates almost non-identifiability of $G_{Ks}$ (Appendix 2).

Larger inaccuracies were found in the inverse problem solutions on experimental data (*Figure 10*, *Table 4*). The first reason may be low parameter identifiability and we want to highlight inaccuracies in estimating the pharmacological parameters related to $G_{Kr}$, $P_{Ca}$, and $G_{NaL}$ when the hERG channel was blocked in parallel to the Cav1.2 channel (verapamil) or in parallel to both the Cav1.2 and the Nav1.5-late channel (terfenadine). The hERG channel block (prolongation of the AP), and the Cav1.2 and Nav1.5-late channel block (shortening of the AP) are known to have opposite effects on the AP *Orvos, 2019*. At the given drug concentrations, these effects were apparently counterbalancing, which resulted in negligible changes of the AP (*Figure 9*). This situation made the estimation of pharmacological parameters very challenging and led to particularly large inaccuracies for terfenadine. The accurate estimations of the pharmacological parameters related to GKs are surprising at first in light of the almost non-identifiability. This was due to the combination of two factors: (1) different from synthetic data, the drugs at the given concentrations did not affect the corresponding KCNQ1-MinK channel and (2) the difference between control and drugged maximum conductance was weakly enforced to be minimal (see Time series fitting and estimation of maximum conductances and pharmacological parameters) which leads to almost no difference in non-identifiable parameters and hence, to a pharmacological parameter of one. The accuracy will likely be much lower in drugs that affect the KCNQ1-MinK channel.

The second and probably main reason for the inaccuracies may be the fact that the data were collected in small tissue preparations, whereas the emulator was trained on data generated by a simulator that represents single cardiomyocytes. APs in small tissue preparations are slightly different from those in single cardiomyocytes. Differences can arise from electrotonic coupling and the mixture of cells including fibroblasts that are able to modify the EP (*Kohl and Gourdie, 2014*; *Mayer et al., 2017*; *Hall et al., 2021*). This can hamper the fitting of the APs and consequently, the estimation of the maximum conductances and pharmacological parameters.

## Emulator

The presented NN emulator enables a massive speed-up compared to regular simulations and the evaluation for the forward problem on synthetic data showed also highly accurate AP emulations. Cardiomyocyte EP models are already very quick to evaluate in the scale of seconds (see Computational performance), but the achieved runtime of emulations allows to solve time consuming simulation protocols markedly more efficient. One such scenario is the presented inverse maximum conductance estimation problem (see Inverse problem based on synthetic data and Inverse problem based on experimental data), where for estimating maximum conductances of a single AP, we need to emulate the steady state AP at least several hundred times as part of an optimization procedure. Further applications include the probabilistic use of cardiomyocyte EP models with uncertainty quantification (*Chang et al., 2017*; *Johnstone et al., 2016*) where thousands of samples of parameters are potentially necessary to compute a distribution of the steady-state properties of subsequent APs, and the creation of cell populations (*Muszkiewicz et al., 2016*; *Gemmell et al., 2016*; *Britton et al., 2013*). In addition to the aforementioned strengths of the presented NN emulator, some further valuable features are worth mentioning that arise from the continuous nature of the emulator. First, the AP can be emulated and fitted at any desired resolution. Second, timing offsets, for example between stimuli in the data to be fitted and the training data, can be accounted for using $t_0$ in (3) without retraining. Last but not least, the transmembrane potential gradient $\frac{dV_m}{dt}$, recently highlighted in terms of proarrhythmic potential prediction (*Jeong et al., 2022*), can be continuously derived and is not dependent on the temporal discretization.

## Limitations and future work

Some limitations have to be considered. First, the emulator has only maximum conductances as inputs. Although these explain much of the AP variability seen between cardiomyocytes (*Britton et al., 2013*; *Muszkiewicz et al., 2016*), the inclusion of parameters related to the channel kinetics might enable a more detailed consideration of drug effects in pharmacological studies. In general, the question of complete parameter identifiability utilizing only APs remains an open challenge (*Zaniboni et al., 2010*). Channel kinetics determine the contribution of the corresponding current to the AP generation in different phases and can thus also modulate drug effects, but were however neglected, as the expansion of the input space might be unsuitable for solving the inverse problem when only AP data are used. Second, the interaction between drugs and their targets is solely captured through scaling of the related maximum conductance at control, which is mostly adequate but in fact an oversimplification (*Brennan et al., 2009*). The interaction can be dependent on time, voltage, and channel state, which requires the use of Markov models with many more parameters (*Brennan et al., 2009*; *Li et al., 2017*; *Lei et al., 2024*). Again, this expands the input space and might be unsuitable for solving the inverse problem when only AP data are used. Moreover, drugs that are applied over a longer period of time can also cause modifications of the maximum conductances through changes in gene expression (*Shim et al., 2023*). This requires attention to avoid misinterpretations of found blocking or enhancement effects, for example by estimating the control maximum conductances again after a washout procedure. Third, the inverse problem was only solved for AP data obtained from one single stimulation protocol. *Johnstone et al., 2016* have shown that the usage of AP data obtained from various stimulation protocols can improve the parameter identifiability and thus, the accuracy of parameter estimates. To be able to use those data in the presented approach, the pacing cycle length must be included as additional input in the emulator and the emulator may be trained on more than the last AP of the pacing series. This would also allow to capture alternans. Last but not least, the number of drugs and concentrations considered in the inverse problem on experimental data poses a limitation. The ultimate goal is to have a tool that provides highly accurate solutions for drugs with different targets and concentrations. To this end, analyses must be extended by data obtained from a series of available and well characterized drugs. The data should be collected in single cardiomyocytes in order to minimize the discussed inaccuracies that stem from the use of tissue preparation data. Further simulation studies of the inverse problem on tissue slabs versus cardiomyocyte EP model could be integrated to assess the impact of differences in setups. This should be addressed in future work. Of note, the presented approach can also be straightforwardly applied to other transients, for example intracellular $[Ca^{2+}]$ or sarcomere length.

## Conclusion

This paper introduced NN emulation of the human ventricular cardiomyocyte AP and tested its applicability in pharmacological studies. The computational cost of the NN emulator was compared to that of the simulator, revealing a massive speed-up of more than $1e^3$. The accuracy of solving the forward problem on synthetic data was found to be high for normal APs and this hold mostly true for abnormal APs exhibiting EADs. This advantage distinguishes our novel approach from existing emulation methods. While larger inaccuracies were observed when utilizing experimental data – a limitation thoroughly discussed and particularly inherent to the fact that small tissue preparations were studied while the emulator was trained on single cardiomyocyte data – the accuracy of solving the inverse problem on synthetic data remained high. Collectively, these findings underscore the potential of NN emulators in improving the efficiency of future quantitative systems pharmacology studies.

## Acknowledgements

This research was funded by the Austrian Science Fund (FWF), the German Research Foundation (DFG), the National Institutes of Health, the National Research, Development and Innovation Office (NRDI), the Hungarian Research Network (HUN-REN), and the Wellcome Trust. For the purpose of open access, the authors have applied a CC-BY public copyright licence to any Author Accepted Manuscript version arising from this submission.

## Additional information

### Funding

| Funder | Grant reference number | Author |
|---|---|---|
| Austrian Science Fund | ERA-NET I 4652-B | Christoph M Augustin |
| National Institutes of Health | R01-HL158667 | Christoph M Augustin |
| German Research Foundation | Walter Benjamin Fellowship No. 468256475 | Alexander Jung |
| HUN-REN Hungarian Research Network | | Norbert Jost András Varró |
| Wellcome Trust | Senior Research Fellowship No. 212203/Z/18/Z | Gary R Mirams |
| National Research Development and Innovation Office | No. 142738 | Norbert Jost András Varró |

The funders had no role in study design, data collection and interpretation, or the decision to submit the work for publication. For the purpose of Open Access, the authors have applied a CC BY public copyright license to any Author Accepted Manuscript version arising from this submission.

### Author contributions

Thomas Grandits, Conceptualization, Data curation, Software, Formal analysis, Validation, Investigation, Visualization, Methodology, Writing - original draft, Writing – review and editing; Christoph M Augustin, Steven A Niederer, Funding acquisition, Writing – review and editing; Gundolf Haase, Gernot Plank, András Varró, László Virág, Resources, Funding acquisition, Writing – review and editing; Norbert Jost, Resources, Formal analysis, Writing – review and editing; Gary R Mirams, Funding acquisition, Methodology, Writing – review and editing; Alexander Jung, Conceptualization, Data curation, Formal analysis, Funding acquisition, Validation, Investigation, Visualization, Writing - original draft, Project administration, Writing – review and editing

### Author ORCIDs

Thomas Grandits ⓘ http://orcid.org/0000-0001-7441-8002
Christoph M Augustin ⓘ http://orcid.org/0000-0001-6341-4014
Gundolf Haase ⓘ http://orcid.org/0000-0002-3439-6117
Gary R Mirams ⓘ https://orcid.org/0000-0002-4569-4312
Steven A Niederer ⓘ http://orcid.org/0000-0002-4612-6982
András Varró ⓘ http://orcid.org/0000-0003-0745-3603
Alexander Jung ⓘ http://orcid.org/0000-0002-0723-4403

### Ethics

Data set #4 contains data of small right ventricular trabeculae and papillary tissue preparations that were obtained from healthy human hearts (see Orvos et al., 2019). Hearts were obtained from organ donors whose hearts were explanted to obtain pulmonary and aortic valves for transplant surgery. Before cardiac explantation, organ donors did not receive medication apart from dobutamine, furosemide, and plasma expanders. Proper consent was obtained for use of each individual's tissue for experimentation. The investigations conform to the principles of the Declaration of Helsinki. Experimental protocols were approved by the University of Szeged and National Scientific and Research Ethical Review Boards (No. 51-57/1997 OEj and 4991-0/2010-1018EKU [339/PI/010]).

Reviewer #1 (Public Review): https://doi.org/10.7554/eLife.91911.3.sa1
Reviewer #3 (Public Review): https://doi.org/10.7554/eLife.91911.3.sa2
Author response https://doi.org/10.7554/eLife.91911.3.sa3

## Additional files

### Supplementary files
• MDAR checklist

### Data availability
The trained emulator is available as a python package from https://github.com/thomgrand/cardiomyocyte_emulator (copy archived at *Thomas, 2024*; the code is licensed under AGPLv3, see https://www.gnu.org/licenses/agpl-3.0.en.html for details). The trained emulator is provided as a Python package, heavily utilizing PyTorch (*Paszke, 2019*) for the neural network execution, allowing it to be executed on both CPUs and NVidia GPUs. Additionally, NumPy (*Harris et al., 2020*) interfaces are provided for easy interfacing with other libraries. Note that the provided repository is a re-implementation of the code-base used in this study and thus may deviate in performance or runtime. The training data set (data set #1) is available on Zenodo using the link https://zenodo.org/records/10640339.

The following dataset was generated:

| Author(s) | Year | Dataset title | Dataset URL | Database and Identifier |
|---|---|---|---|---|
| Grandits T, Augustin C, Haase G, Jost N, Mirams G, Niederer S, Plank G, Varró V, Virág L, Jung A | 2024 | Cardiomyocyte Emulator Training Data | https:/doi.org/10.5281/zenodo.10640339 | Zenodo, 10.5281/zenodo.10640339 |

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

## Appendix 1

### AP biomarkers and abnormalities

The AP biomarkers used in this study were selected such that the key characteristics of the depolarization and the repolarization phase can be quantified. They include $RMP$ (resting transmembrane potential measured just before stimulation), $dVmMax$ (maximum transmembrane potential slope during the upstroke), $Peak$ (peak transmembrane potential at the end of the upstroke), $TP$ (time-to-peak from the stimulus until $Peak$ is reached), $APD_x$ (AP duration at $x \in \{30, 40, 50, 60, 70, 80, 90\}$% repolarization relative to the AP amplitude ($Peak - RMP$) measured from the instant of $dVmMax$), and $Tri_{90-40}$ (triangulation defined as the difference between $APD_{90}$ and $APD_{40}$ **Britton et al., 2017**).

For the GSA (see.3) and the creation of synthetic data (see Synthetic test data (#2/#3)), AP abnormalities were also considered which included depolarization abnormalities, repolarization abnormalities, and alternans. Depolarization abnormalities were defined as an upstroke peak below 0 mV and an AP that does not reach 0 mV before 100 ms after stimulation (**Passini et al., 2017**). Repolarization abnormalities were defined as a transmembrane potential rate of rise of more than $0.01 \frac{\text{mV}}{\text{ms}}$ from 150 ms after the upstroke peak onwards (representative of early afterdepolarizations) and as a transmembrane potential that does not fall below -40 mV (**Passini et al., 2017**) (representative of repolarization failure). Alternans were defined as $APD_{90}$ difference of more than 5 ms between two consecutive APs (**Morotti et al., 2021**).

## Appendix 2

## Global sensitivity analysis

A variance-based Sobol' global sensitivity analysis (GSA) (*Sobol', 2001*) was performed on the simulator to quantify the sensitivities of the maximum channel conductances (inputs) with respect to the AP biomarkers (outputs; see AP biomarkers and abnormalities). This informed the decision on which inputs to consider in the emulator. Furthermore, it was used for the interpretation of the solutions of the inverse problem since parameters that are insensitive with respect to the outputs indicate non-identifiability (*Guillaume et al., 2019*).

The maximum conductances used for building the model population in *Tomek et al., 2019* were considered and Saltelli's sampling scheme (*Saltelli, 2002*) was applied with $N = 1024$ to generate 20,480 input samples with values between 50% and 150% of the original values. Simulations were performed for each input sample and biomarker values (see AP biomarkers and abnormalities) derived from the last AP were used for the analysis. However, data were excluded if not all biomarkers could be determined or abnormalities (see AP biomarkers and abnormalities) were detected in the last two consecutive APs. First-order (S1) and total-effect (ST) Sobol' sensitivity indices were computed using the Saltelli method (*Homma and Saltelli, 1996*; *Saltelli, 2002*). This requires outputs for each input sample and to take this into account, excluded outputs were assigned the mean values of included outputs. The `SALib-Sensitivity Analysis Library` (*Herman and Usher, 2017*) was used for the GSA.

The GSA could be performed on the data of all input samples as no data were excluded. S1 and ST were mostly very similar which indicates only little interactions among the maximum conductivities relative to the AP biomarkers (9). The only exception was $TP$. As was to be expected, the analysis underlines the predominant relative sensitivity of $G_{Na}$ with respect to biomarkers of the depolarization phase, the predominant relative sensitivity of $G_{NaL}$, $G_{Kr}$, and $G_{NCX}$ with respect to biomarkers of the repolarization phase and the predominant relative sensitivity of $G_{K1}$, $P_{NaK}$ to the resting transmembrane potential. However, $G_{Ks}$ has a negligible relative sensitivity to all biomarkers. This indicates almost non-identifiability.

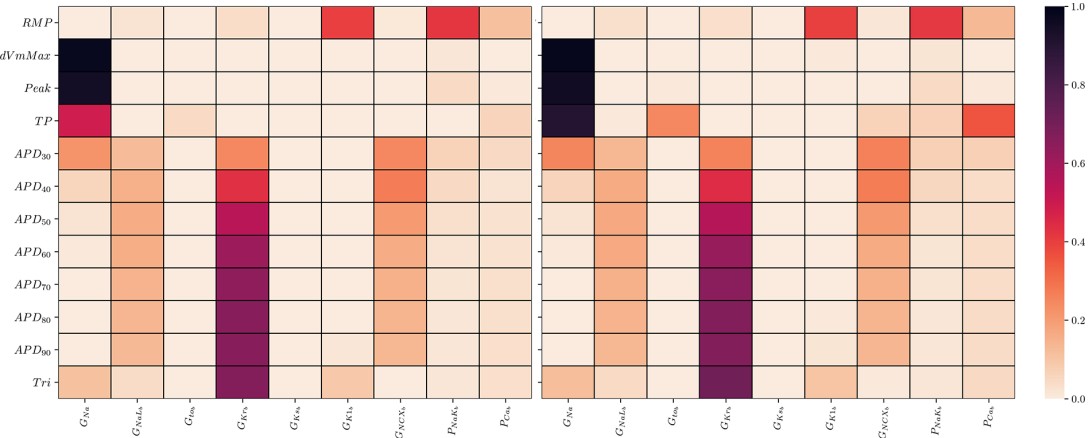

**Appendix 2—figure 1.** Global sensitivity analysis of the ToR-ORd simulator. Sobol' sensitivity indices are shown for each maximum conductance relative to each AP biomarker. Left: first-order (S1), right: total-effect (ST) Sobol' sensitivity coefficient.

## Appendix 3

## EAD classification

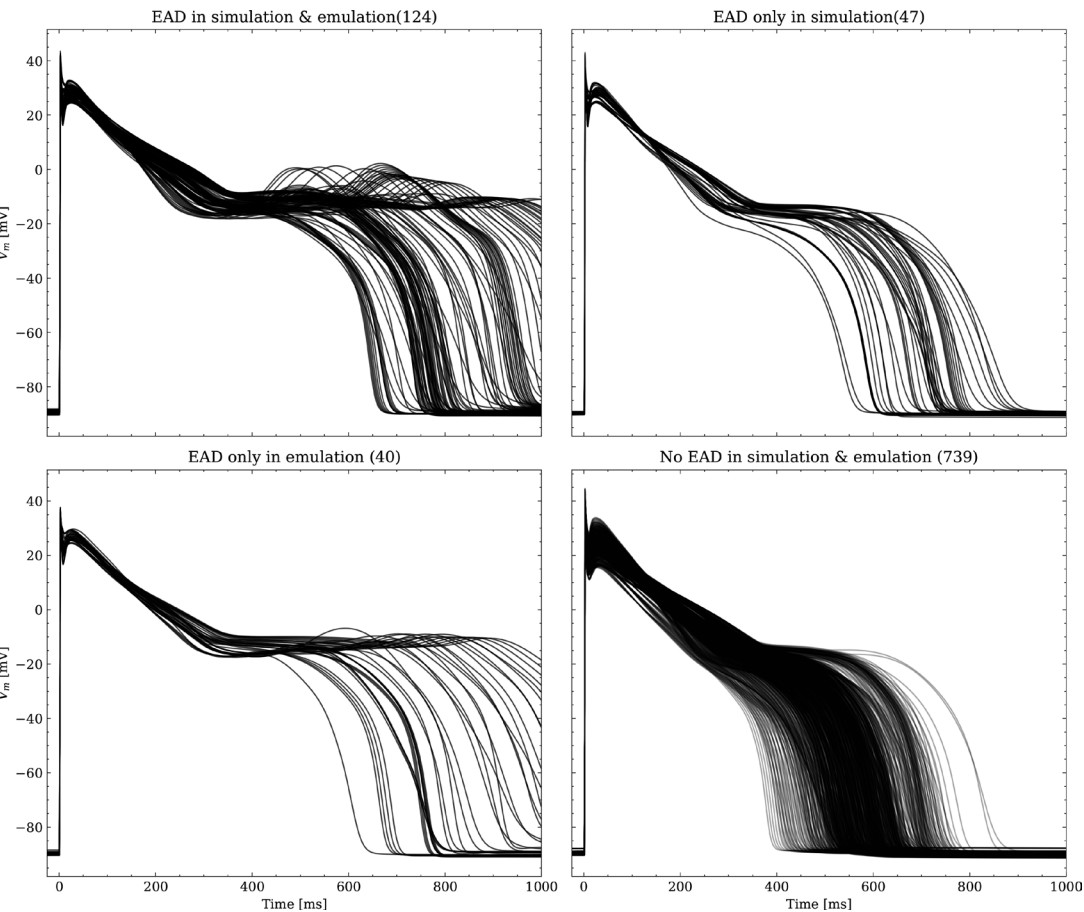

**Appendix 3—figure 1.** Emulated APs based on the pharmacological parameters of data set #3. See also Forward problem and **Figure 7**. From left to right and top to bottom, the plot shows the true positive, false negative, false positive and true negative samples. The number next to the title specifies the number of samples belonging to each category. The classification criterion is outlined in Appendix 1.

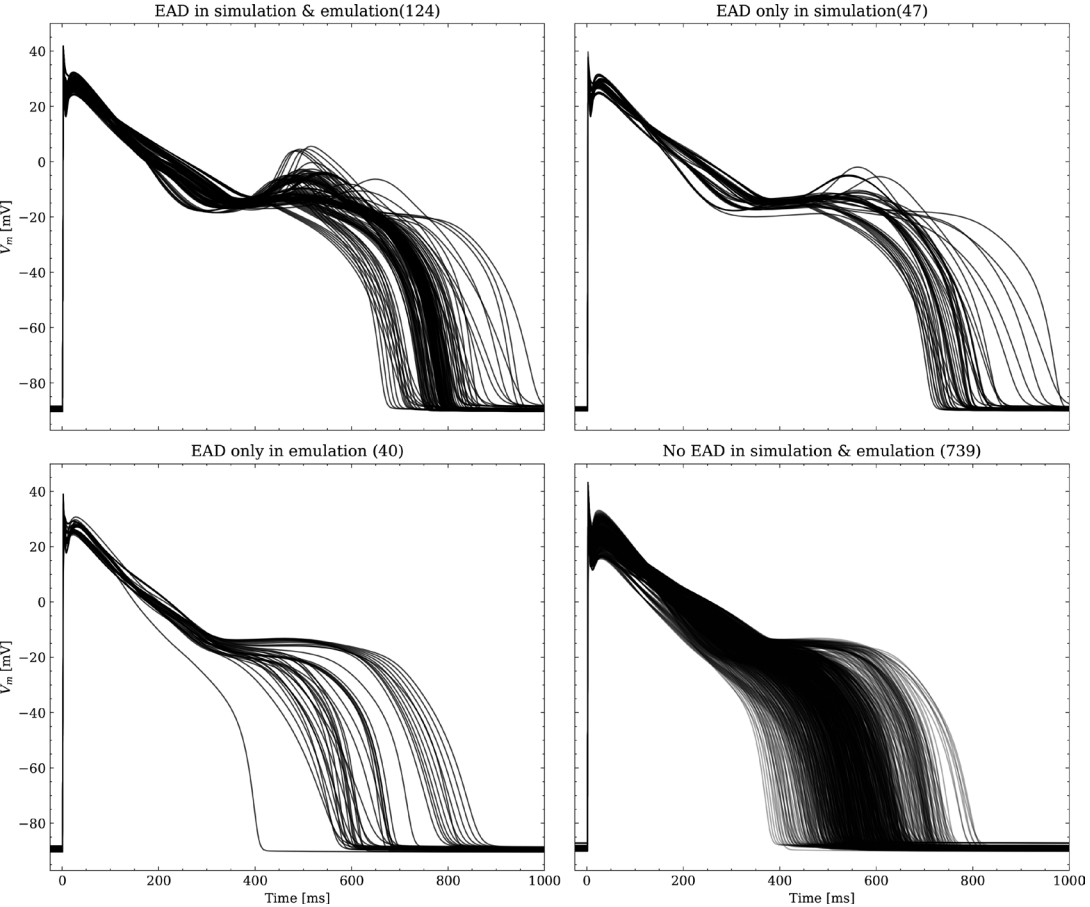

**Appendix 3—figure 2.** Same as *Figure 1*, but showing the simulated APs.

