## [Editor Report · eLife assessment]

This **valuable** prospective study develops a new tool to accelerate pharmacological studies by using neural networks to emulate the human ventricular cardiomyocyte action potential. The evidence supporting the conclusions is **convincing**, based on using a large and high-quality dataset to train the neural network emulator. There are nevertheless a few areas in which the article may be improved through validating the neural network emulators against extensive experimental data. In addition, the article may be improved through delineating the exact speed-up achieved and the scope for acceleration. The work will be of broad interest to scientists working in cardiac simulation and quantitative system pharmacology.

---

## [Referee Report · Reviewer #1 (Public Review)]

Summary:

The authors present a neural network (NN)-based approach to computationally cheaper emulation of simulations of biophysically relatively detailed cardiac cell models based on systems of ordinary differential equations. Relevant case studies are used to demonstrate the performance in prediction of standard action potentials, as well as action potentials manifesting early depolarizations. Application to the "reverse problem" (inferring the effect of pharmacological compounds on ion channels based on action potential data before and after drug treatment) is also explored, which is a task of generally high interest.

Strengths:

This is a well-designed study, which explores an area that many in the cardiac simulation community will be interested in. The article is well written and I particularly commend the authors on transparency of methods description, code sharing, etc. - it feels rather exemplary in this regard and I only wish more authors of cardiac simulation studies took such an approach. The training speed of the network is encouraging and the technique is accessible to anyone with a reasonably strong GPU, not needing specialized equipment.

Weaknesses:

Below are several points that I consider to be weaknesses and/or uncertainties of the work:

1. The scope for acceleration of single cell simulations is not vast, as it is easy to simulate tens of thousands of cells per day on a workstation computer, using simulation conditions similar to those of the authors. While this covers a large part of what is needed in the field, I agree with the authors that there are applications where the presented technology is helpful. In such cases, e.g., in uncertaintly quantification, it will enable studies that would be difficult to carry out previously. In addition, any application involving long-term pre-pacing of a large number of cells will benefit greatly from the reported tool.

An area which is definitely in need of acceleration is simulations of whole ventricles or hearts, but it is not clear how much potential for speedup would the presented technology bring there. I can imagine interesting applications of rapid emulation in such a setting, some of which could be hybrid in nature (e.g. using simulation for the region around the wavefront of propagating electrical waves, while emulating the rest of the tissue, which is behaving more regularly/predictable, and is likely to be emulated well), but this is definitely beyond of the scope of this article.

2. The exact speed-up achieved by the NN emulation is somewhat context-dependent. In particular, the reported speedup critically depends on the number of beats in the simulation. The emulator learns to directly estimate the state of the cell after X beats (where X is decided by the operator of training). The speedup appears to be relatively marginal when a single beat is simulated versus emulated - but when 1000 beats are simulated, this takes 1000fold more time for simulation, but unchanged time for emulation.

While the initial submission did not communicate the practical speedup entirely clearly, this was addressed well by the authors in the revised version.

3. It appears that the accuracy of emulation drops off relatively sharply with increasing real-world applicability/relevance of the tasks it is applied to. That said, the authors are to be commended on declaring this transparently, rather than withholding such analyses. I particularly enjoyed the discussion of the not always amazing results of the inverse problem on the experimental data. The point on low parameter identifiability is an important one, and serves as a warning against overconfidence in our ability to infer cellular parameters from action potentials alone. On the other hand, I'm not that sure the difference between small tissue preps and single cells which authors propose as another source of the discrepancy will be that vast beyond the AP peak potential (probably much of the tissue prep is affected by the pacing electrode?), but that is a subjective view only. The influence of coupling could be checked if the simulated data were generated from 2D tissue samples/fibres, e.g. using the Myokit software.

In summary, I believe the range of tasks where the emulator provides a major advance is relatively narrow, particularly given the relatively limited need for further speedup compared to simulations. However, this does not make the study uninteresting in the slightest - on the contrary, it explores something that many of us are thinking about, and it is likely to stimulate further development in the direction of computationally efficient emulation of relatively complex simulations.

---

## [Referee Report · Reviewer #3 (Public Review)]

Summary:

1. Grandits and colleagues were trying to develop a new tool to accelerate pharmacological studies by using neural networks to emulate the human ventricular cardiomyocyte action potential (AP). The AP is a complex electrical signal that governs the heartbeat, and it is important to accurately model the effects of drugs on the AP to assess their safety and efficacy. Traditional biophysical simulations of the AP are computationally expensive and time-consuming. The authors hypothesized that neural network emulators could be trained to predict the AP with high accuracy and that these emulators could also be used to quickly and accurately predict the effects of drugs on the AP.

Strengths:

2. One of the study's major strengths is that the authors use a large and high-quality dataset to train their neural network emulator. The dataset includes a wide range of APs, including normal and abnormal APs exhibiting EADs. This ensures that the emulator is robust and can be used to predict the AP for a variety of different conditions.

Another major strength of the study is that the authors demonstrate that their neural network emulator can be used to accelerate pharmacological studies. For example, they use the emulator to predict the effects of a set of known arrhythmogenic drugs on the AP. The emulator is able to predict the effects of these drugs, even though it had not been trained on these drugs specifically.

Weaknesses:

One weakness of the study is that it is important to validate neural network emulators against experimental data to ensure that they are accurate and reliable. The authors do this to some extent, but further validation would be beneficial. In particular for the inverse problem, where the estimation of pharmacological parameters very challenging and led to particularly large inaccuracies.

Additional context:

4. The work by Grandits et al. has the potential to revolutionize the way that pharmacological studies are conducted. Neural network emulation has the promise to reduce the time and cost of drug development and to improve the safety and efficacy of new drugs. The methods and data presented in the paper are useful to the community because they provide a starting point for other researchers to develop and improve neural network emulators for the human ventricular cardiomyocyte AP. The authors have made their code and data publicly available, which will facilitate further research in this area.

5. It is important to note that neural network emulation is still a relatively new approach, and there are some challenges that need to be addressed before it can be widely adopted in the pharmaceutical industry. For example, neural network emulators need to be trained on large and high-quality datasets. Additionally, it is important to validate neural network emulators against experimental data to ensure that they are accurate and reliable. Despite these challenges, the potential benefits of neural network emulation for pharmacological studies are significant. As neural network emulation technology continues to develop, it is likely to become a valuable tool for drug discovery and development.

---

## [Author Response]

The following is the authors’ response to the original reviews.

**eLife assessment**
This valuable study reports on the potential of neural networks to emulate simulations of human ventricular cardiomyocyte action potentials for various ion channel parameters with the advantage of saving simulation time in certain conditions. The evidence supporting the claims of the authors is solid, although the inclusion of open analysis of drop-off accuracy and validation of the neural network emulators against experimental data would have strengthened the study. The work will be of interest to scientists working in cardiac simulation and quantitative pharmacology.

Thank you for the kind assessment. It is important for us to point out that, while limited, experimental validation was performed in this study and is thoroughly described in the work.

**Reviewer 1 - Comments**
This manuscript describes a method to solve the inverse problem of finding the initial cardiac activations to produce a desired ECG. This is an important question. The techniques presented are novel and clearly demonstrate that they work in the given situation. The paper is well-organized and logical.Strengths:This is a well-designed study, which explores an area that many in the cardiac simulation community will be interested in. The article is well written and I particularly commend the authors on transparency of methods description, code sharing, etc. - it feels rather exemplary in this regard and I only wish more authors of cardiac simulation studies took such an approach. The training speed of the network is encouraging and the technique is accessible to anyone with a reasonably strong GPU, not needing specialized equipment.Weaknesses:Below are several points that I consider to be weaknesses and/or uncertainties of the work:C I-(a) I am not convinced by the authors’ premise that there is a great need for further acceleration of cellular cardiac simulations - it is easy to simulate tens of thousands of cells per day on a workstation computer, using simulation conditions similar to those of the authors. I do not really see an unsolved task in the field that would require further speedup of single-cell simulations.At the same time, simulations offer multiple advantages, such as the possibility to dissect mechanisms of the model behaviour, and the capability to test its behaviour in a wide array of protocols - whereas a NN is trained for a single purpose/protocol, and does not enable a deep investigation of mechanisms. Therefore, I am not sure the cost/benefit ratio is that strong for single-cell emulation currently.An area that is definitely in need of acceleration is simulations of whole ventricles or hearts, but it is not clear how much potential for speedup the presented technology would bring there. I can imagine interesting applications of rapid emulation in such a setting, some of which could be hybrid in nature (e.g. using simulation for the region around the wavefront of propagating electrical waves, while emulating the rest of the tissue, which is behaving more regularly/predictable, and is likely to be emulated well), but this is definitely beyond of the scope of this article.

Thank you for this point of view. Simulating a population of few thousand cells is completely feasible on single desktop machines and for fixed, known parameters, emulation may not fill ones need. Yet we still foresee a great untapped potential for rapid evaluations of ionic models, such as for the gradient-based inverse problem, presented in the paper. Such inverse optimization requires several thousand evaluations per cell and thus finding maximum conductances for the presented experimental data set (13 cell pairs control/drug → 26 APs) purely through simulations would require roughly a day of simulation time even in a very conservative estimation (3.5 seconds per simulation, 1000 simulations per optimization). Additionally, the emulator provides local sensitivity information between the AP and maximum conductances in the form of the gradient, which enables a whole new array of efficient optimization algorithms [Beck, 2017]. To further emphasize these points, we added the number of emulations and runtime of each conducted experiment in the specific section and a paragraph in the discussion that addresses this point:

"Cardiomyocyte EP models are already very quick to evaluate in the scale of seconds (see Section 2.3.1), but the achieved runtime of emulations allows to solve time consuming simulation protocols markedly more efficient. One such scenario is the presented inverse maximum conductance estimation problem (see Section 3.1.2 and Section 3.1.3), where for estimating maximum conductances of a single AP, we need to emulate the steady state AP at least several hundred times as part of an optimization procedure. Further applications include the probabilistic use of cardiomyocyte EP models with uncertainty quantification [Chang et al., 2017, Johnstone et al., 2016] where thousands of samples of parameters are potentially necessary to compute a distribution of the steady-state properties of subsequent APs, and the creation of cell populations [Muszkiewicz et al., 2016, Gemmell et al., 2016, Britton et al., 2013]." (Section 4.2)

We believe that rapid emulations are valuable for several use-cases, where thousands of evaluations are necessary. These include the shown inverse problem, but similarly arise in uncertainty quantification, or cardiomyocyte population creation. Similarly, new use-cases may arise as such efficient tools become available. Additionally, we provided the number of evaluations along with the runtimes for each of the conducted experiments, showing how essential these speedups are to realize these experiments in reasonable timeframes. Utilizing these emulations in organ-level electrophysiological models is a possibility, but the potential problems in such scenarios are much more varied and depend on a number of factors, making it hard to pin-point the achievable speed-up using ionic emulations.

C I-(b) The authors run a cell simulation for 1000 beats, training the NN emulator to mimic the last beat. It is reported that the simulation of a single cell takes 293 seconds, while emulation takes only milliseconds, implying a massive speedup. However, I consider the claimed speedup achieved by emulation to be highly context-dependent, and somewhat too flattering to the presented method of emulation. Two specific points below:First, it appears that a not overly efficient (fixed-step) numerical solver scheme is used for the simulation. On my (comparable, also a Threadripper) CPU, using the same model (”ToR-ORd-dyncl”), but a variable step solver ode15s in Matlab, a simulation of a cell for 1000 beats takes ca. 50 seconds, rather than 293 of the authors. This can be further sped up by parallelization when more cells than available cores are simulated: on 32 cores, this translates into ca. 2 seconds amortized time per cell simulation (I suspect that the NN-based approach cannot be parallelized in a similar way?). By amortization, I mean that if 32 models can be simulated at once, a simulation of X cells will not take X*50 seconds, but (X/32)*50. (with only minor overhead, as this task scales well across cores).Second, and this is perhaps more important - the reported speed-up critically depends on the number of beats in the simulation - if I am reading the article correctly, the runtime compares a simulation of 1000 beats versus the emulation of a single beat. If I run a simulation of a single beat across multiple simulated cells (on a 32-core machine), the amortized runtime is around 20 ms per cell, which is only marginally slower than the NN emulation. On the other hand, if the model was simulated for aeons, comparing this to a fixed runtime of the NN, one can get an arbitrarily high speedup.Therefore, I’d probably emphasize the concrete speedup less in an abstract and I’d provide some background on the speedup calculation such as above, so that the readers understand the context-dependence. That said, I do think that a simulation for anywhere between 250 and 1000 beats is among the most reasonable points of comparison (long enough for reasonable stability, but not too long to beat an already stable horse; pun with stables was actually completely unintended, but here it is...). I.e., the speedup observed is still valuable and valid, albeit in (I believe) a somewhat limited sense.

We agree that the speedup comparison only focused on a very specific case and needs to be more thoroughly discussed and benchmarked. One of the main strengths of the emulator is to cut the time of prepacing to steady state, which is known to be a potential bottleneck for the speed of the single-cell simulations. The time it takes to reach the steady state in the simulator is heavily dependant on the actual maximum conductance configuration and the speed-up is thus heavily reliant on a per-case basis. The differences in architecture of the simulator and emulator further makes direct comparisons very difficult. In the revised version we now go into more detail regarding the runtime calculations and also compare it to an adaptive time stepping simulation (Myokit [Clerx et al., 2016]) in a new subsection:

"The simulation of a single AP (see Section 2.1) sampled at a resolution of 20kHz took 293s on one core of a AMD Ryzen Threadripper 2990WX (clock rate: 3.0GHz) in CARPentry. Adaptive timestep solver of variable order, such as implemented in Myokit [Clerx et al., 2016], can significantly lower the simulation time (30s for our setup) by using small step sizes close to the depolarization (phase 0) and increasing the time step in all other phases. The emulation of a steady state AP sampled at a resolution of 20kHz for t ∈ [−10, 1000]ms took 18.7ms on a AMD Ryzen 7 3800X (clock rate: 3.9GHz) and 1.2ms on a Nvidia A100 (Nvidia Corporation, USA), including synchronization and data copy overhead between CPU and GPU.

"The amount of required beats to reach the steady state of the cell in the simulator has a major impact on the runtime and is not known a-priori. On the other hand, both simulator and emulator runtime linearly depends on the time resolution, but since the output of the emulator is learned, the time resolution can be chosen at arbitrarily without affecting the AP at the sampled times. This makes direct performance comparisons between the two methodologies difficult. To still be able to quantify the speed-up, we ran Myokit using 100 beats to reach steady state, taking 3.2s of simulation time. In this scenario, we witnessed a speed-up of 171 and 2 · 103 of our emulator on CPU and GPU respectively (again including synchronization and data copy overhead between CPU and GPU in the latter case). Note that both methods are similarly expected to have a linear parallelization speedup across multiple cells.

For the inverse problem, we parallelized the problem for multiple cells and keep the problem on the GPU to minimize the overhead, achieving emulations (including backpropagation) that run in 120µs per AP at an average temporal resolution of 2kHz. We consider this the peak performance which will be necessary for the inverse problem in Section 3.1.2." (Section 2.3.1)

Note that the mentioned parallelization across multiple machines/hardware applies equally to the emulator and simulator (linear speed-up), though the utilization for single cells is most likely different (single vs. multi-cell parallelization).

C I-(c) It appears that the accuracy of emulation drops off relatively sharply with increasing real-world applicability/relevance of the tasks it is applied to. That said, the authors are to be commended on declaring this transparently, rather than withholding such analyses. I particularly enjoyed the discussion of the not-always amazing results of the inverse problem on the experimental data. The point on low parameter identifiability is an important one and serves as a warning against overconfidence in our ability to infer cellular parameters from action potentials alone. On the other hand, I’m not that sure the difference between small tissue preps and single cells which authors propose as another source of the discrepancy will be that vast beyond the AP peak potential (probably much of the tissue prep is affected by the pacing electrode?), but that is a subjective view only. The influence of coupling could be checked if the simulated data were generated from 2D tissue samples/fibres, e.g. using the Myokit software.Given the points above (particularly the uncertain need for further speedup compared to running single-cell simulations), I am not sure that the technology generated will be that broadly adopted in the near future.However, this does not make the study uninteresting in the slightest - on the contrary, it explores something that many of us are thinking about, and it is likely to stimulate further development in the direction of computationally efficient emulation of relatively complex simulations.

We agree that the parameter identifiability is an important point of discussion. While the provided experimental data gave us great insights already, we still believe that given the differences in the setup, we can not draw conclusions about the source of inaccuracies with absolute certainty. The suggested experiment to test the influence of coupling is of interest for future works and has been integrated into the discussion. Further details are given in the response to the recommendation R III- (t)

**Reviewer 2 - Comments**
Summary:This study provided a neural network emulator of the human ventricular cardiomyocyte action potential. The inputs are the corresponding maximum conductances and the output is the action potential (AP). It used the forward and inverse problems to evaluate the model. The forward problem was solved for synthetic data, while the inverse problem was solved for both synthetic and experimental data. The NN emulator tool enables the acceleration of simulations, maintains high accuracy in modeling APs, effectively handles experimental data, and enhances the overall efficiency of pharmacological studies. This, in turn, has the potential to advance drug development and safety assessment in the field of cardiac electrophysiology.Strengths:1. Low computational cost: The NN emulator demonstrated a massive speed-up of more than 10,000 times compared to the simulator. This substantial increase in computational speed has the potential to expedite research and drug development processes1. High accuracy in the forward problem: The NN emulator exhibited high accuracy in solving the forward problem when tested with synthetic data. It accurately predicted normal APs and, to a large extent, abnormal APs with early afterdepolarizations (EADs). High accuracy is a notable advantage over existing emulation methods, as it ensures reliable modeling and prediction of AP behaviorC II-(a) Input space constraints: The emulator relies on maximum conductances as inputs, which explain a significant portion of the AP variability between cardiomyocytes. Expanding the input space to include channel kinetics parameters might be challenging when solving the inverse problem with only AP data available.

Thank you for this comment. We consider this limitation a major drawback, as discussed in Section 4.3. Identifiability is already an issue when only considering the most important maximum conductances. Further extending the problem to include kinetics will most likely only increase the difficulty of the inverse problem. For the forward problem though, it might be of interest to people studying ionic models to further analyze the effects of channel kinetics.

C II-(b) Simplified drug-target interaction: In reality, drug interactions can be time-, voltage-, and channel statedependent, requiring more complex models with multiple parameters compared to the oversimplified model that represents the drug-target interactions by scaling the maximum conductance at control. The complex model could also pose challenges when solving the inverse problem using only AP data.

Thank you pointing out this limitation. We slightly adapted Section 4.3 to further highlight some of these limitations. Note however that the experimental drugs used have been shown to be influenced by this drug interaction in varying degrees [Li et al., 2017] (e.g. dofetilide vs. cisapride). However, the discrepancy in identifiability was mostly channel-based (0%-100%), whereas the variation in identifiability between drugs was much lower (39%-66%).

C II-(c) Limited data variety: The inverse problem was solved using AP data obtained from a single stimulation protocol, potentially limiting the accuracy of parameter estimates. Including AP data from various stimulation protocols and incorporating pacing cycle length as an additional input could improve parameter identifiability and the accuracy of predictions.

The proposed emulator architecture currently only considers the discussed maximum conductances as input and thus can only compensate when using different stimulation protocols. However, the architecture itself does not prohibit including any of these as parameters for future variants of the emulator. We potentially foresee future works extending on the architecture with modified datasets to include other parameters of importance, such as channel kinetics, stimulation protocols and pacing cycle lengths. These will however vary between the actual use-cases one is interested in.

C II-(d) Larger inaccuracies in the inverse problem using experimental data: The reasons for this result are not quite clear. Hypotheses suggest that it may be attributed to the low parameter identifiability or the training data set were collected in small tissue preparation.

The low parameter identifiability on some channels (e.g. GK1) poses a problem, for which we state multiple potential reasons. As of yet, no final conclusion can be drawn, warranting further research in this area.

**Reviewer 3 - Comments**
Summary:Grandits and colleagues were trying to develop a new tool to accelerate pharmacological studies by using neural networks to emulate the human ventricular cardiomyocyte action potential (AP). The AP is a complex electrical signal that governs the heartbeat, and it is important to accurately model the effects of drugs on the AP to assess their safety and efficacy. Traditional biophysical simulations of the AP are computationally expensive and time-consuming. The authors hypothesized that neural network emulators could be trained to predict the AP with high accuracy and that these emulators could also be used to quickly and accurately predict the effects of drugs on the AP.Strengths:One of the study’s major strengths is that the authors use a large and high-quality dataset to train their neural network emulator. The dataset includes a wide range of APs, including normal and abnormal APs exhibiting EADs. This ensures that the emulator is robust and can be used to predict the AP for a variety of different conditions.Another major strength of the study is that the authors demonstrate that their neural network emulator can be used to accelerate pharmacological studies. For example, they use the emulator to predict the effects of a set of known arrhythmogenic drugs on the AP. The emulator is able to predict the effects of these drugs, even though it had not been trained on these drugs specifically.C III-(a) One weakness of the study is that it is important to validate neural network emulators against experimental data to ensure that they are accurate and reliable. The authors do this to some extent, but further validation would be beneficial. In particular for the inverse problem, where the estimation of pharmacological parameters was very challenging and led to particularly large inaccuracies.

Thank you for this recommendation. Further experimental validation of the emulator in the context of the inverse problem would be definitely beneficial. Still, an important observation is that the identifiability varies greatly between channels. While the inverse problem is an essential reason for utilizing the emulator, it is also empirically validated for the pure forward problem and synthetic inverse problem, together with the (limited) experimental validation. The sources of problems arising in estimating the maximum conductances of the experimental tissue preparations are important to discuss in future works, as we now further emphasize in the discussion. See also the response to the recommendations R III-(t).

**Reviewer 1 - Recommendations**
R I-(a) Could further detail on the software used for the emulation be provided? E.g. based on section 2.2.2, it sounds like a CPU, as well as GPU-based emulation, is possible, which is neat.

Indeed as suspected, the emulator can run on both CPUs and GPUs and features automatic parallelization (per-cell, but also multi-cell), which is enabled by the engineering feats of PyTorch [Paszke et al., 2019]. This is now outlined in a bit more detail in Sec. 2 and 5.

"The trained emulator is provided as a Python package, heavily utilizing PyTorch [Paszke et al., 2019] for the neural network execution, allowing it to be executed on both CPUs and NVidia GPUs." (Section 5)

R I-(b) I believe that a potential use of NN emulation could be also in helping save time on prepacing models to stability - using the NN for ”rough” prepacing (e.g. 1000 beats), and then running a simulation from that point for a smaller amount of time (e.g. 50 beats). One could monitor the stability of states, so if the prepacing was inaccurate, one could quickly tell that these models develop their state vector substantially, and they should be simulated for longer for full accuracy - but if the model was stable within the 50 simulated beats, it could be kept as it is. In this way, the speedup of the NN and accuracy and insightfulness of the simulation could be combined. However, as I mentioned in the public review, I’m not sure there is a great need for further speedup of single-cell simulations. Such a hybrid scheme as described above might be perhaps used to accelerate genetic algorithms used to develop new models, where it’s true that hundreds of thousands to millions of cells are eventually simulated, and a speedup there could be practical. However one would have to have a separate NN trained for each protocol in the fitness function that is to be accelerated, and this would have to be retrained for each explored model architecture. I’m not sure if the extra effort would be worth it - but maybe yes to some people.

Thank you for this valuable suggestion. As pointed out in C I-(a), one goal of this study was to reduce the timeconsuming task of prepacing. Still, in its current form the emulator could not be utilized for prepacing simulators, as only the AP is computed by the emulator. For initializing a simulation at the N-th beat, one would additionally need all computed channel state variables. However, a simple adaptation of the emulator architecture would allow to also output the mentioned state variables.

R I-(c) Re: ”Several emulator architectures were tried on the training and validation data sets and the final choice was hand-picked as a good trade-off between high accuracy and low computational cost” - is it that the emulator architecture was chosen early in the development, and the analyses presented in the paper were all done with one previously selected architecture? Or is it that the analyses were attempted with all considered architectures, and the well-performing one was chosen? In the latter case, this could flatter the performance artificially and a test set evaluation would be worth carrying out.

We apologize for the unclear description of the architectural validation. The validation was in fact carried out with 20% of the training data (data set #1), which is however completely disjoint with the test set (#2, #3, #4, formerly data set #1 and #2) on which the evaluation was presented. To further clarify the four different data sets used in the study, we now dedicated an additional section to describing each set and where it was used (see also our response below R I-(d)), and summarize them in Table 1, which we also added at R II-(a). The cited statement was slightly reworked.

"Several emulator architectures were tried on the training and validation data sets and the final choice was hand-picked as a good trade-off between high accuracy on the validation set (#1) and low computational runtime cost." (Section 2.2.2)

R I-(d) When using synthetic data for the forward and inverse problem, with the various simulated drugs, is it that split of the data into training/validation test set was done by the drug simulated (i.e., putting 80 drugs and the underlying models in the training set, and 20 into test set)? Or were the data all mixed together, and 20% (including drugs in the test set) were used for validation? I’m slightly concerned by the potential of ”soft” data leaks between training/validation sets if the latter holds. Presumably, the real-world use case, especially for the inverse problem, will be to test drugs that were not seen in any form in the training process. I’m also not sure whether it’s okay to reuse cell models (sets of max conductances) between training and validation tests - wouldn’t it be better if these were also entirely distinct? Could you please comment on this?

We completely agree with the main points of apprehension that training, validation and test sets all serve a distinct purpose and should not be arbitrarily mixed. However, this is only a result of the sub-optimal description of our datasets, which we heavily revised in Section 2.2.1 (Data, formerly 2.3.1). We now present the data using four distinct numbers: The initial training/validation data, now called data set #1 (formerly no number), is split 80%/20% into training and validation sets (for architectural choices) respectively. The presented evaluations in Section 2.3 (Evaluation) are purely performed on data set #2 (normal APs, formerly #1), #3 (EADs, formerly #2) and #4 (experimental).

R I-(e) For the forward problem on EADs, I’m not sure if the 72% accuracy is that great (although I do agree that the traces in Fig 12-left also typically show substantial ICaL reactivation, but this definitely should be present, given the IKr and ICaL changes). I would suggest that you also consider the following design for the EAD investigation: include models with less severe upregulation of ICaL and downregulation of IKr, getting a population of models where a part manifests EADs and a part does not. Then you could run the emulator on the input data of this population and be able to quantify true, falsexpositive, negative detections. I think this is closer to a real-world use case where we have drug parameters and a cell population, and we want to quickly assess the arrhythmic risk, with some drugs being likely entirely nonrisky, some entirely risky, and some between (although I still am not convinced it’s that much of an issue to just simulate this in a couple of thousands of cells).

Thank you for pointing out this alternative to address the EAD identification task. Even though the values chosen in Table 2 seem excessively large, we still only witnessed EADs in 171 of the 950 samples. Especially border cases, which are close to exhibiting EADs are hardest to estimate for the NN emulator. As suggested, we now include the study with the full 950 samples (non-EAD & EAD) and classify the emulator AP into one of the labels for each sample. The mentioned 72.5% now represent the sensitivity, whereas our accuracy in such a scenario becomes90.8% (total ratio of correct classifications):

"The data set #3 was used second and Appendix C shows all emulated APs, both containing the EAD and non-EAD cases. The emulation of all 950 APs took 0.76s on the GPU specified in Section 2.2.3 We show the emulation of all maximum conductances and the classification of the emulation. The comparison with the actual EAD classification (based on the criterion outlined in Appendix A) results in true-positive (EAD both in the simulation and emulation), false-negative (EAD in the simulation, but not in the emulation), false-positive (EAD in the emulation, but not in the simulation) and true-negative (no EAD both in the emulation and simulation). The emulations achieved 72.5% sensitivity (EAD cases correctly classified) and 94.9% specificity (non-EAD cases correctly classified), with an overall accuracy of 90.8% (total samples correctly classified). A substantial amount of wrongly classified APs showcase a notable proximity to the threshold of manifesting EADs. Figure 7 illustrates the distribution of RMSEs in the EAD APs between emulated and ground truth drugged APs. The average RMSE over all EAD APs was 14.5mV with 37.1mV being the maximum. Largest mismatches were located in phase 3 of the AP, in particular in emulated APs that did not fully repolarize." (Section 3.1.1)

R I-(f) Figure 1 - I think a large number of readers will understand the mathematical notation describing inputs/outputs; that said, there may be a substantial number of readers who may find that hard to read (e.g. lab-based researchers, or simulation-based researchers not familiar with machine learning). At the same time, this is a very important part of the paper to explain what is done where, so I wonder whether using words to describe the inputs/outputs would not be more practical and easier to understand (e.g. ”drug-based conductance scaling factor” instead of ”s” ?). It’s just an idea - it needs to be tried to see if it wouldn’t make the figure too cluttered.

We agree that the mathematical notation may be confusing to some readers. As a compromise between using verbose wording and mathematical notation, we introduced a legend in the lower right corner of the figure that shortly describes the notation in order to help with interpreting the figure.

R I-(g) ”APs with a transmembrane potential difference of more than 10% of the amplitude between t = 0 and 1000 ms were excluded” - I’m not sure I understand what exactly you mean here - could you clarify?

With this criterion, we try to discard data that is far away from fully repolarizing within the given time frame, which applies to 116 APs in data set #1 and 50 APs in data set #3. We added a small side note into the text:

"APs with a transmembrane potential difference of more than 10% of the amplitude between t = 0 and 1000ms (indicative of an AP that is far away from full repolarization) were excluded." (Section 2.2.1)

R I-(h) Speculation (for the future) - it looks like a tool like this could be equally well used to predict current traces, as well as action potentials. I wonder, would there be a likely benefit in feeding back the currents-traces predictions on the input of the AP predictor to provide additional information? Then again, this might be already encoded within the network - not sure.

Although not possible with the chosen architecture (see also R I-(b)), it is worth thinking about an implementation in future works and to study differences to the current emulator.

Entirely minor points:R I-(i) ”principle component analysis” → principal component analysis

Fixed

R I-(j) The paper will be probably typeset by elife anyway, but the figures are often quite far from their sections, with Results figures even overflowing into Discussion. This can be often fixed by using the !htb parameters (\begin{figure}[!htb]), or potentially by using ”\usepackage[section]{placeins}” and then ”\FloatBarrier” at the start and end of each section (or subsection) - this prevents floating objects from passing such barriers.

Thank you for these helpful suggestions. We tried reducing the spacing between the figures and their references in the text, hopefully improving the reader’s experience.

R I-(k) Alternans seems to be defined in Appendix A (as well as repo-/depolarization abnormalities), but is not really investigated. Or are you defining these just for the purpose of explaining what sorts of data were also included in the data?

We defined alternans since this was an exclusion criterion for generating simulation data.

**Reviewer 2 - Recommendations**
R II-(a) Justification for methods selection: Explain the rationale behind important choices, such as the selection of specific parameters and algorithms.

Thank you for this recommendation, we tried to increase transparency of our choices by introducing a separate data section that summarizes all data sets and their use cases in Section 2.2.1 and also collect many of the explanations there. Additionally we added an overview table (Table 1) of the utilized data.

**Author response table 1. sa3table1:** Summary of the data used in this study, along with their usage and the number of valid samples. Note that each AP is counted individually, also in cases of control/drug pairs.

ID	Description	Usage	Origin	Samples
#1	Training/validation data	Training and validating the emulator, choosing the best architecture (Section ‘Architecture’)	Simulation	39,884
#2	Synthetic drug data, normal APs	Testing forward and inverse performance for normal APs (Sections ‘Forward problem’ and ‘Inverse problem based on synthetic data’)	Simulation	10^4^
#3	Synthetic drug data, including EAD APs	Testing forward performance of abnormal (EAD) APs (‘Forward problem’)	Simulation	950
#4	Experimental cardiomyocytes	Testing and comparing the inverse performance with data published by the CiPA initiative (Li et al.. 2017, Chang et al., 2017; Section ‘Inverse problem based onexperimental data’)	Orvos et al., 2019	26

R II-(b) Interpretation of the evaluation results: After presenting the evaluation results, consider interpretations or insights into what the results mean for the performance of the emulator. Explain whether the emulator achieved the desired accuracy or compare it with other existing methods.In the revised version, we tried to further expand the discussion on possible applications of our emulator (Section 4.2). See also our response to C I-(a). To the best of our knowledge, there are currently no out-of-the-box methods available for directly comparing all experiments we considered in our work.
**Reviewer 3 - Recommendations**
R III-(a) In the introduction (Page 3) and then also in the 2.1 paragraph authors speak about the ”limit cycle”: Do you mean steady state conditions? In that case, it is more common to use steady state.

When speaking about the limit cycle, we refer to what is also sometimes called the steady state, depending on the field of research and/or personal preference. We now mention both terms at the first occurence, but stick with the limit cycle terminology which can also be found in other works, see e.g. [Endresen and Skarland, 2000].

R III-(b) On page 3, while comparing NN with GP emulators, I still don’t understand the key reason why NN can solve the discontinuous functions with more precision than GP.

The potential problems in modeling sharp continuities using GPs is further explained in the referenced work [Ghosh et al., 2018] and further references therein:

"Statistical emulators such as Gaussian processes are frequently used to reduce the computational cost of uncertainty quantification, but discontinuities render a standard Gaussian process emulation approach unsuitable as these emulators assume a smooth and continuous response to changes in parameter values [...] Applying GPs to model discontinuous functions is largely an open problem. Although many advances (see the discussion about non-stationarity in [Shahriari et al., 2016] and the references in there) have been made towards solving this problem, a common solution has not yet emerged. In the recent GP literature there are two specific streams of work that have been proposed for modelling non-stationary response surfaces including those with discontinuities. The first approach is based on designing nonstationary processes [Snoek et al., 2014] whereas the other approach attempts to divide the input space into separate regions and build separate GP models for each of the segmented regions. [...]"([Ghosh et al., 2018])

We integrated a short segment of this explanation into Section 1.

R III-(c) Why do authors prefer to use CARPentry and not directly openCARP?The use of CARPentry is purely a practical choice since the simulation pipeline was already set up. As we now point out however in Sec. 2.1 (Simulator), simulations can also be performed using any openly available ionic simulation tool, such as Myokit [Clerx et al., 2016], OpenCOR [Garny and Hunter, 2015] and openCARP [Plank et al., 2021]. We emphasized this in the text.

"Note, that the simulations can also be performed using open-source software such as Myokit [Clerx et al., 2016], OpenCOR [Garny and Hunter, 2015] and openCARP [Plank et al., 2021]." (Section 2.1)

R III-(d) In paragraph 2.1:(a) In this sentence: ”Various solver and sampling time steps were applied to generate APs and the biomarkers used in this study (see Appendix A)” this reviewer suggests putting the Appendix reference near “biomarkers”. In addition, a figure that shows the test of various solver vs. sampling time steps could be interesting and can be added to the Appendix as well.(b) Why did the authors set the relative difference below 5% for all biomarkers? Please give a reference to that choice. Instead, why choose 2% for the time step?

1. We adjusted the reference to be closer to “biomarkers”. While we agree that further details on the influence of the sampling step would be of interest to some of the readers, we feel that it is far beyond the scope of this paper.

1. There is no specific reference we can provide for the choice. Our goal was to reach 5% relative difference, which we surpassed by the chosen time steps of 0.01 ms (solver) and 0.05 ms (sampling), leading to only 2% difference. We rephrased the sentence in question to make this clear.

"We considered the time steps with only 2% relative difference for all AP biomarkers (solver: 0.01ms; sampling: 0.05ms) to offer a sufficiently good approximation." (Section 2.1)

R III-(e) In the caption of Figure 1 authors should include the reference for AP experimental data (are they from Orvos et al. 2019 as reported in the Experimental Data section?)

We added the missing reference as requested. As correctly assumed, they are from [Orvos et al., 2019].

R III-(f) Why do authors not use experimental data in the emulator development/training?

For the supervised training of our NN emulator, we need to provide the maximum conductances of our chosen channels for each AP. While it would be beneficial to also include experimental data in the training to diversify the training data, the exact maximum conductances in our the considered retrospective experiments are not known. In the case such data would be available with low measurement uncertainty, it would be possible to include.

R III-(g) What is TP used in the Appendix B? I could not find the acronymous explanation.

We are sorry for the oversight, TP refers to the time-to-peak and is now described in Appendix A.

R III-(h) Are there any reasons for only using ST and no S1? Maybe are the same?

The global sensitivity analysis is further outlined in Appendix B, also showing S1 (first-order effects) and ST (variance of all interactions) together (Figure 11) [Herman and Usher, 2017] and their differences (e.g. in TP) Since S1 only captures first-order effects, it may fail to capture higher-order interactions between the maximum conductances, thus we favored ST.

R III-(i) In Training Section Page 8. It is not clear why it is necessary to resample data. Can you motivate?

The resampling part is motivated by exactly capturing the swift depolarization dynamics, whereas the output from CARPentry is uniformly sampled. This is now further highlighted in the text.

"Then, the data were non-uniformly resampled from the original uniformly simulated APs, to emphasize the depolarization slope with a high accuracy while lowering the number of repolarization samples. For this purpose, we resamled the APs [...]" (Section 2.2.1)

R III-(j) For the training of the neuronal network, the authors used the ADAM algorithm: have you tested any other algorithm?

For training neural networks, ADAM has become the current de-facto standard and is certainly a robust choice for training our emulator. While there may exist slightly faster, or better-suited training algorithms, we witnessed (qualitative) convergence in the training (Equation (2)). We thus strongly believe that the training algorithm is not a limiting factor in our study.

R III-(k) What is the amount of the drugs tested? Is the same dose reported in the description of the second data set or the values are only referring to experimental data? Moreover, it is not clear if in the description of experimental data, the authors are referring to newly acquired data (since they described in detail the protocol) or if they are obtained from Orvos et al. 2019 work.

In all scenarios, we tested 5 different drugs (cisapride, dofetilide, sotalol, terfenadine, verapamil). We revised our previous presentation of the data available, and now try to give a concise overview over the utilized data (Section 2.2.1 and table 1) and drug comparison with the CiPA distributions (Table 5, former 4). Note that in the latter case, the available expected channel scaling factors by the CiPA distributions vary, but are now clearly shown in Table 5.

R III-(l) In Figure 4, I will avoid the use of “control” in the legend since it is commonly associated with basal conditions and not with the drug administration.

The terminology “control” in this context is in line with works from the CiPA initiative, e.g. [Li et al., 2017] and refers to the state of cell conditions before the drug wash-in. We added a minor note the first time we use the term control in the introduction to emphasize that we refer to the state of the cell before administering any drugs

"To compute the drugged AP for given pharmacological parameters is a forward problem, while the corresponding inverse problem is to find pharmacological parameters for given control (before drug administration) and drugged AP." (Section 1)

R III-(m) In Table 1 when you referred to Britton et al. 2017 work, I suggest adding also 10.1371/journal.pcbi.1002061.

We added the suggested article as a reference.

R III-(n) For the minimization problem, only data set #1 has been used. Have you tested data set #2?

In the current scenario, we only tested the inverse problem for data set #2 (former #1). The main purpose for data set #3 (former #2), was to test the possibility to emulate EAD APs. Given the overall lower performance in comparison to data set #2 (former #1), we also expect deteriorated results in comparison to the existing inverse synthetic problem.

R III-(o) In Figure 6 you should have the same x-axis (we could not see any points in the large time scale for many biomarkers). Why dVmMax is not uniformed distributed compared to the others? Can you comment on that?

As suggested, we re-adjusted the x-range to show the center of distributions. Additionally, we denoted in each subplot the number of outliers which lie outside of the shown range. The error distribution on dVmMax exhibits a slightly off-center, left-tailed normal distribution, which we now describe a bit more in the revised text:

"While the mismatches in phase 3 were simply a result of imperfect emulation, the mismatches in phase 0 were a result of the difficulty in matching the depolarization time exactly. [...] Likewise, the difficulty in exactly matching the depolarization time leads to elevated errors and more outliers in the biomarkers influenced by the depolarization phase (TP and dVmMax)," (Section 3.1.1)

R III-(p) Page 14. Can the authors better clarify ”the average RMSE over all APs 13.6mV”: is it the mean for all histograms in Figure 7? (In Figure 5 is more evident the average RMSE).

The average RMSE uses the same definition for Figures 5 and 7: It is the average over all the RMSEs for each pair of traces (simulated/emulated), though the amount of samples is much lower for the EAD data set and not normal distributed.

R III-(q) In Table 4, the information on which drugs are considered should be added.For each channel, we added the names of the drugs for which respective data from the CiPA initiative were available.R III-(r) Pag. 18, second paragraph, there is a repetition of ”and”.

Fixed

R III-(s) The pair’s combination of scaling factors for simulating synthetic drugs reported in Table 2, can be associated with some effects of real drugs? In this case, I suggest including the information or justifying the choice.

The scaling factors in Table 2 are used to create data set #3 (former #2), and is meant to provide several APs which expose EADs. This is described in more detail in the new data section, Section 2.2.1:

"Data set #3: The motivation for creating data set #3 was to test the emulator on data of abnormal APs showing the repolarization abnormality EAD. This is considered a particularly relevant AP abnormality in pharmacological studies because of their role in the genesis of drug-induced ventricular arrhythmia’s [Weiss et al., 2010]. Drug data were created using ten synthetic drugs with the hERG channel and the Cav1.2 channel as targets. To this end, ten samples with pharmacological parameters for GKr and PCa (Table 2) were generated and the synthetic drugs were applied to the entire synthetic cardiomyocyte population by scaling GKr and PCa with the corresponding pharmacological parameter. Of the 1000 APs simulated, we discarded APs with a transmembrane potential difference of more than 10% of the amplitude between t = 0 and 1000ms (checked for the last AP), indicative of an AP that does not repolarize within 1000ms. This left us with 950 APs, 171 of which exhibit EAD (see Appendix C)." (Section 2.2.1)

R III-(t) A general comment on the work is that the authors claim that their study highlights the potential of NN emulators as a powerful tool for increased efficiency in future quantitative systems pharmacology studies, but they wrote ”Larger inaccuracies were found in the inverse problem solutions on experimental data highlight inaccuracies in estimating the pharmacological parameters”: so, I was wondering how they can claim the robustness of NN use as a tool for more efficient computation in pharmacological studies.

The discussed robustness directly refers to efficiently emulating steady-state/limit cycle APs from a set of maximum conductances (forward problem, Section 3.1.1). We extensively evaluated the algorithm and feel that given the low emulation RMSE of APs (< 1 mV), the statement is warranted. The inverse estimation, enabled through this rapid evaluation, performs well on synthetic data, but shows difficulties for experimental data. Note however that at this point there are multiple potential sources for these problems as highlighted in the Evaluation section (Section 4.1) and Table 5 (former 4) highlights the difference in accuracy of estimating per-channel maximum conductances, revealing a potentially large discrepancy. The emulator also offers future possibilities to incorporate additional informations in the forms of either priors, or more detailed measurements (e.g. calcium transients) and can be potentially improved to a point where also the inverse problem can be satisfactorily solved in experimental preparations, though further analysis will be required.

References[Beck, 2017] Beck, A. (2017). First-order methods in optimization. SIAM.

[Britton et al., 2013] Britton, O. J., Bueno-Orovio, A., Ammel, K. V., Lu, H. R., Towart, R., Gallacher, D. J., and Rodriguez, B. (2013). Experimentally calibrated population of models predicts and explains intersubject variability in cardiac cellular electrophysiology. Proceedings of the National Academy of Sciences, 110(23).

[Chang et al., 2017] Chang, K. C., Dutta, S., Mirams, G. R., Beattie, K. A., Sheng, J., Tran, P. N., Wu, M., Wu, W. W., Colatsky, T., Strauss, D. G., and Li, Z. (2017). Uncertainty quantification reveals the importance of data variability and experimental design considerations for in silico proarrhythmia risk assessment. Frontiers in Physiology, 8.

[Clerx et al., 2016] Clerx, M., Collins, P., de Lange, E., and Volders, P. G. A. (2016). Myokit: A simple interface to cardiac cellular electrophysiology. Progress in Biophysics and Molecular Biology, 120(1):100–114.

[Endresen and Skarland, 2000] Endresen, L. and Skarland, N. (2000). Limit cycle oscillations in pacemaker cells. IEEE Transactions on Biomedical Engineering, 47(8):1134–1137.

[Garny and Hunter, 2015] Garny, A. and Hunter, P. J. (2015). OpenCOR: a modular and interoperable approach to computational biology. Frontiers in Physiology, 6.

[Gemmell et al., 2016] Gemmell, P., Burrage, K., Rodr´ıguez, B., and Quinn, T. A. (2016). Rabbit-specific computational modelling of ventricular cell electrophysiology: Using populations of models to explore variability in the response to ischemia. Progress in Biophysics and Molecular Biology, 121(2):169–184.

[Ghosh et al., 2018] Ghosh, S., Gavaghan, D. J., and Mirams, G. R. (2018). Gaussian process emulation for discontinuous response surfaces with applications for cardiac electrophysiology models.

[Herman and Usher, 2017] Herman, J. and Usher, W. (2017). SALib: An open-source python library for sensitivity analysis. J. Open Source Softw., 2(9):97.

[Johnstone et al., 2016] Johnstone, R. H., Chang, E. T., Bardenet, R., de Boer, T. P., Gavaghan, D. J., Pathmanathan, P., Clayton, R. H., and Mirams, G. R. (2016). Uncertainty and variability in models of the cardiac action potential: Can we build trustworthy models? Journal of Molecular and Cellular Cardiology, 96:49–62.

[Li et al., 2017] Li, Z., Dutta, S., Sheng, J., Tran, P. N., Wu, W., Chang, K., Mdluli, T., Strauss, D. G., and Colatsky, T. (2017). Improving the in silico assessment of proarrhythmia risk by combining hERG (human ether`a-go-go-related gene) channel–drug binding kinetics and multichannel pharmacology. Circulation: Arrhythmia and Electrophysiology, 10(2).

[Muszkiewicz et al., 2016] Muszkiewicz, A., Britton, O. J., Gemmell, P., Passini, E., S´anchez, C., Zhou, X., Carusi, A., Quinn, T. A., Burrage, K., Bueno-Orovio, A., and Rodriguez, B. (2016). Variability in cardiac electrophysiology: Using experimentally-calibrated populations of models to move beyond the single virtual physiological human paradigm. Progress in Biophysics and Molecular Biology, 120(1):115–127.

[Orvos et al., 2019] Orvos, P., Kohajda, Z., Szlov´ak, J., Gazdag, P., Arp´adffy-Lovas, T., T´oth, D., Geramipour, A.,´ T´alosi, L., Jost, N., Varr´o, A., and Vir´ag, L. (2019). Evaluation of possible proarrhythmic potency: Comparison of the effect of dofetilide, cisapride, sotalol, terfenadine, and verapamil on hERG and native iKr currents and on cardiac action potential. Toxicological Sciences, 168(2):365–380.

[Paszke et al., 2019] Paszke, A., Gross, S., Massa, F., Lerer, A., Bradbury, J., Chanan, G., Killeen, T., Lin, Z., Gimelshein, N., Antiga, L., Desmaison, A., Kopf, A., Yang, E., DeVito, Z., Raison, M., Tejani, A., Chilamkurthy, S., Steiner, B., Fang, L., Bai, J., and Chintala, S. (2019). PyTorch: An Imperative Style, High-Performance Deep Learning Library. In Advances in Neural Information Processing Systems, volume 32. Curran Associates, Inc.

[Plank et al., 2021] Plank, G., Loewe, A., Neic, A., Augustin, C., Huang, Y.-L., Gsell, M. A., Karabelas, E., Nothstein, M., Prassl, A. J., S´anchez, J., Seemann, G., and Vigmond, E. J. (2021). The openCARP simulation environment for cardiac electrophysiology. Computer Methods and Programs in Biomedicine, 208:106223.

[Shahriari et al., 2016] Shahriari, B., Swersky, K., Wang, Z., Adams, R. P., and de Freitas, N. (2016). Taking the Human Out of the Loop: A Review of Bayesian Optimization. Proceedings of the IEEE, 104(1):148–175. Conference Name: Proceedings of the IEEE.

[Snoek et al., 2014] Snoek, J., Swersky, K., Zemel, R., and Adams, R. (2014). Input Warping for Bayesian Optimization of Non-Stationary Functions. In Proceedings of the 31st International Conference on Machine Learning, pages 1674–1682. PMLR. ISSN: 1938-7228.

[Weiss et al., 2010] Weiss, J. N., Garfinkel, A., Karagueuzian, H. S., Chen, P.-S., and Qu, Z. (2010). Early afterdepolarizations and cardiac arrhythmias. Heart Rhythm, 7(12):1891–1899.